# Unlearnable 3D Point Clouds: Class-wise Transformation Is All You Need

**Xianlong Wang**[1,2,4,5†], **Minghui Li**[‡,*], **Wei Liu**[1,2,4,5†], **Hangtao Zhang**[4,5†],
**Shengshan Hu**[1,2,4,5†], **Yechao Zhang**[1,2,4,5†], **Ziqi Zhou**[1,2,3§], **Hai Jin**[1,2,3§]

[1] National Engineering Research Center for Big Data Technology and System
[2] Services Computing Technology and System Lab   [3] Cluster and Grid Computing Lab
[4] Hubei Engineering Research Center on Big Data Security
[5] Hubei Key Laboratory of Distributed System Security
† School of Cyber Science and Engineering, Huazhong University of Science and Technology
‡ School of Software Engineering, Huazhong University of Science and Technology
§ School of Computer Science and Technology, Huazhong University of Science and Technology
`{wxl99,minghuili,weiliu73,hangt_zhang,hushengshan,ycz,zhouziqi,hjin}@hust.edu.cn`

## Abstract

Traditional unlearnable strategies have been proposed to prevent unauthorized users from training on the 2D image data. With more 3D point cloud data containing sensitivity information, unauthorized usage of this new type data has also become a serious concern. To address this, we propose the first integral unlearnable framework for 3D point clouds including two processes: (i) we propose an unlearnable data protection scheme, involving a class-wise setting established by a category-adaptive allocation strategy and multi-transformations assigned to samples; (ii) we propose a data restoration scheme that utilizes class-wise inverse matrix transformation, thus enabling authorized-only training for unlearnable data. This restoration process is a practical issue overlooked in most existing unlearnable literature, *i.e.*, even authorized users struggle to gain knowledge from 3D unlearnable data. Both theoretical and empirical results (including 6 datasets, 16 models, and 2 tasks) demonstrate the effectiveness of our proposed unlearnable framework. Our code is available at https://github.com/CGCL-codes/UnlearnablePC.

## 1   Introduction

Recently, 3D point cloud deep learning has been making remarkable strides in various domains, *e.g.*, self-driving [6] and virtual reality [1, 46]. Specifically, numerous 3D sensors scan the surrounding environment and synthesize massive 3D point cloud data containing sensitive information such as pedestrian and vehicles [32] to the cloud server for deep learning analysis [12, 23]. However, the raw point cloud data can be exploited for point cloud unauthorized deep learning if a data breach occurs, posing a significant privacy threat. Fortunately, the privacy protection approaches for preventing unauthorized training have been extensively studied in the 2D image domain [9, 19, 28, 29, 39, 42]. They apply elaborate perturbations on images such that trained networks over them exhibit extremely low generalization, thus failing to learn knowledge from the protected data, known as "making data unlearnable". Nonetheless, the stark disparity between 2D images and 3D point clouds poses significant challenges for drawing lessons from existing 2D solutions.

Specifically, migrating 2D unlearnable schemes to 3D suffers from following challenges: **(i) Incompatibility with 3D data.** Numerous model-agnostic 2D image unlearnable schemes operate

---

*Minghui Li is the corresponding author.

| Transformations | None | Rotation* | Scaling* | Shear | Twisting | Tapering | Reflection* | Translation* |
|---|---|---|---|---|---|---|---|---|
| Illustrations of 3D point cloud samples | | | | | | | | |
| Descriptions | Clean sample | Changing the angles | Changing the size | Stretching or compressing | Twisting the shape | Narrowing or shortening | Creating a mirror object | Changing the position |
| Is it reversible? | — | ✓ | ✓ | ✓ | ✓ | ✕ | ✓ | ✓ |
| Quantity of possible transformations | — | Infinite | Infinite | Infinite | Infinite | Infinite | 3 | Infinite |
| Dimension of the multiplicative transformation matrix | — | $\mathbb{R}^{3\times3}$ | $\mathbb{R}^{3\times3}$ | $\mathbb{R}^{3\times3}$ | $\mathbb{R}^{3\times3}$ | $\mathbb{R}^{3\times3}$ | $\mathbb{R}^{3\times3}$ | $\mathbb{R}^{4\times4}$ |

Figure 1: An overview of existing seven types of 3D transformations. "*" denotes *rigid transformations* that do not alter the shape of the point cloud samples, while the remaining transformations are non-rigid transformations.

in the pixel space, such as convolutional operations [39, 42], making them fail to be directly transferred to the 3D point space. **(ii) Poor visual quality.** Migrating model-dependent 2D unlearnable methods [5, 9, 19, 28] to 3D point clouds requires perturbing substantial points, leading to irregular three-dimensional shifts which may significantly degrade visual quality. Hence, these challenges spur us to start directly from the characteristics of point clouds for proposing 3D unlearnable solutions.

Recent works observe that 3D transformations can alter test-time results of models [7, 17, 44]. To explore this, we conduct an in-depth investigation into the properties of seven 3D transformations as shown in Fig. 1 and reveal the mechanisms by which transformations employed in a certain pattern serve as unlearnable schemes (Sec. 3.2). In light of this, we propose the first unlearnable approach in 3D point clouds via multi class-wise transformation (UMT), transforming samples to various forms for privacy protection. Concretely, we newly propose a category-adaptive allocation strategy by leveraging uniform distribution sampling and category constraints to establish a class-wise setting, thereby multiplying multi-transformations to samples based on categories. To theoretically analyze UMT, we define a binary classification setup similar to that used in [20, 33, 39]. Meanwhile, we employ a *Gaussian Mixture Model* (GMM) [38] to model the clean training set and use the Bayesian optimal decision boundary to model the point cloud classifier. Theoretically, we prove that there exists a UMT training set follows a GMM distribution and the classification accuracy of UMT dataset is lower than that of the clean dataset in a Bayesian classifier.

Moreover, an incompatible issue in existing unlearnable works [9, 19, 28, 29, 39, 43] is identified [54], *i.e.*, these approaches prevent unauthorized learning to protected data, but they also impede authorized users from effectively learning from unlearnable data. To address this, we propose a data restoration scheme that applies class-wise inverse transformations, determined by a lightweight message received from the protector. Our proposed unlearnable framework including UMT approach and data restoration scheme is depicted in Fig. 3.

Extensive experiments on 6 benchmark datasets (including synthetic and real-world datasets) using 16 point cloud models across CNN, MLP, Graph-based Network, and Transformer on two tasks (classification and semantic segmentation), verified the effectiveness of our proposed unlearnable scheme. We summarize our main contributions as follows:

- **The First Integral 3D Unlearnable Framework.** To the best of our knowledge, we propose the first integral unlearnable 3D point cloud framework, utilizing class-wise multi-transformation as its unlearnable mechanism (effectively safeguarding point cloud data against unauthorized exploitation) and proposing a novel data restoration approach that leverages class-wise reversible 3D transformation matrices (addressing an incompatible issue in most existing unlearnable works, where even authorized users cannot effectively learn knowledge from unlearnable data).

- **Theoretical Analysis.** We theoretically indicate the existence of an unlearnable situation that the classification accuracy of the UMT dataset is lower than that of the clean dataset under the decision boundary of the Bayes classifier in Gaussian Mixture Model.

- **Experimental Evaluation.** Extensive experiments on 3 synthetic datasets and 3 real-world datasets using 16 widely used point cloud model architectures on classification and semantic segmentation tasks verify the superiority of our proposed schemes.

## 2 Preliminaries

**Notation.** Considering the raw point cloud data $(\mathbf{X}, \mathbf{Y}) \in \mathcal{X} \times \mathcal{Y}$ sampled from a clean distribution $\mathcal{D}$ for training a point cloud network, the user's goal is to obtain a model $\mathcal{F} : \mathcal{X} \to \mathcal{Y}$ by minimizing the loss function (*e.g.*, cross-entropy loss) $\mathcal{L}(\mathcal{F}(\mathbf{X}), \mathbf{Y})$. Let $\mathbf{T} \in \mathcal{T}$ be a 3D transformation matrix that does not seriously damage the visual quality of point clouds. Note that $\mathcal{X} \in \mathbb{R}^{3 \times p}$, $\mathcal{T} \in \mathbb{R}^{3 \times 3}$, and $p$ represents the number of points. In theoretical analysis, following [20, 33], we simplify a training dataset $\mathcal{D}_k$ to a *Gaussian Mixture Model* (GMM) [38] $\mathcal{N}(y\mu, \boldsymbol{I})$, where $y \in \{\pm 1\}$ denotes the class labels, $\mu \in \mathbb{R}^d$ denotes the mean value, and $\boldsymbol{I} \in \mathbb{R}^{d \times d}$ denotes the identity matrix. Thus the Bayes optimal decision boundary for classifying $\mathcal{D}_k$ is defined by $P(x) \equiv \mu^T x = 0$. The accuracy of the decision boundary $P$ on $\mathcal{D}_k$ is equal to $\phi(\|\mu\|_2)$, where $\phi$ denotes the *Cumulative Distribution Function* (CDF) of the standard normal distribution.

**Data protector $\mathcal{G}_p$.** $\mathcal{G}_p$ aims to protect the knowledge from the clean training set (with size of $n$) $\mathcal{D}_c = \{\mathbf{X}_i, \mathbf{Y}_i\}_{i=1}^n \sim \mathcal{D}$ by compromising the unauthorized models who train on the unlearnable point cloud data $\{\mathbf{T}_i(\mathbf{X}_i), \mathbf{Y}_i\}_{i=1}^n$, resulting in extremely poor generalization on the clean test distribution $\mathcal{D}_t \subseteq \mathcal{D}$. This objective can be formalized as:

$$\max \ \mathbb{E}_{(\mathbf{X},\mathbf{Y}) \sim \mathcal{D}_t} \mathcal{L}\left(\mathcal{F}\left(\mathbf{X}; \theta_u\right), \mathbf{Y}\right), \ \text{s.t.} \ \theta_u = \arg\min_\theta \sum_{(\mathbf{X}_i, \mathbf{Y}_i) \in \mathcal{D}_c} \mathcal{L}\left(\mathcal{F}\left(\mathbf{T}_i(\mathbf{X}_i); \theta\right), \mathbf{Y}_i\right) \quad (1)$$

where $\mathcal{G}_p$ assumes that training samples are all transformed into unlearnable ones while maintaining normal visual effects, in line with previous unlearnable works [19, 28, 39, 42]. By the way, solving Eq. (1) directly is infeasible for neural networks because it necessitates unrolling the entire training procedure within the inner objective and performing backpropagation through it to execute a single step of gradient descent on the outer objective [8].

**Authorized user $\mathcal{G}_a$.** $\mathcal{G}_a$ aims to apply another transformation $\mathbf{T}' \in \mathcal{T}$ on the unlearnable sample, making the protected data learnable. This is formally defined as:

$$\min \ \mathbb{E}_{(\mathbf{X},\mathbf{Y}) \sim \mathcal{D}_t} \mathcal{L}\left(\mathcal{F}\left(\mathbf{X}; \theta_r\right), \mathbf{Y}\right), \ \text{s.t.} \ \theta_r = \arg\min_\theta \sum_{(\mathbf{X}_i, \mathbf{Y}_i) \in \mathcal{D}_c} \mathcal{L}\left(\mathcal{F}\left(\mathbf{T}'_i(\mathbf{T}_i(\mathbf{X}_i)); \theta\right), \mathbf{Y}_i\right) \quad (2)$$

where $\mathcal{G}_a$ assumes that, without access to any clean training samples, $\mathbf{T}'$ can be constructed by utilizing a lightweight message $M$ received from data protectors.

## 3 Our Proposed Unlearnable Schemes

### 3.1 Key Intuition

Several recent works [7, 13, 44] reveal employing 3D transformations can mislead the model's classification results. Such a phenomenon implies that there might be some defects in point cloud classifiers when processing transformed samples, leading us to infer that 3D transformations are probable candidates for data protection against unauthorized training. If the transformed point cloud data are used to train unauthorized DNNs, only simple linear features inherent in 3D transformations (at which transformations may act as shortcuts [14]) are captured by the DNNs, successfully protecting point cloud data privacy.

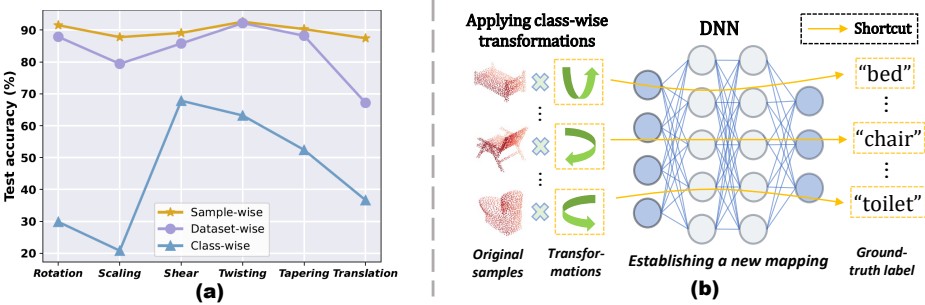

Figure 2: (a) Training on the transformed ModelNet10 dataset (employing sample-wise, dataset-wise, and class-wise patterns) using PointNet classifier; (b) The high-level overview of the class-wise setting

## 3.2 Exploring the Mechanism

We summarize existing 3D transformations in Fig. 1 and formally define them in Appendix A. To seek clarity on the application and selection of transformations, we explore three aspects: **(i) execution mechanism**, **(ii) exclusion mechanism**, and **(iii) working mechanism** as follows.

**(i) Which execution mechanism successfully satisfy Eq. (1)?** The extensively employed execution patterns in 2D unlearnable approaches are sample-wise [9, 19] and class-wise [19, 39] settings. We further complement the dataset-wise setting (using universal transformation) and implement the above execution mechanisms for training a PointNet classifier [35] on the transformed ModelNet10 [50], obtaining test accuracy results in Fig. 2 (a). We discover that model achieves considerably low test accuracy under the class-wise setting, satisfying Eq. (1). Sample-wise and dataset-wise settings do not obviously compromise model performance, which cannot serve as promising unlearnable routes. Moreover, we note that sample-wise transformation is often considered as a data augmentation scheme to improve generalization, which contradicts our aim of using class-wise transformation to lower model generalization.

**(ii) Which transformations need to be excluded?** Not all transformations are suitable candidates. We exclude three transformations, tapering, reflection, and translation. (1) The tapering matrix may cause point cloud samples to become a planar projection when $\eta z$ defined in Eq. (18) equals to -1, rendering the tapered samples meaningless; (2) The reflection matrix has only three distinct transformation matrices, rendering it incapable of assigning class-wise transformations when the number of categories exceeds three; (3) The translation transformation is a straightforward and simple additive transformation that is too easily defeated by point cloud data pre-processing approaches.

**(iii) Why does class-wise transformation work?** We conduct experiments using class-wise transformations (see Tab. 6), indicating that the model training on the class-wise transformed training set achieves a relatively high accuracy on the class-wise transformed test set (using the same transformation process as the training set). Besides, if we permute the class-wise transformation for the test set, we obtain a significant low accuracy on the test set. Therefore, we conclude that the reason why class-wise transformation works is that the model learns the mapping between class-wise transformations and corresponding category labels as shown in Fig. 2 (b), which results in the model being unable to predict the corresponding labels on a clean test set lacking transformations. This analytical process yields conclusions that are in agreement with prior research [39, 44].

## 3.3 Our Design for UMT

### 3.3.1 Category-Adaptive Allocation Strategy

We assign transformation parameters based on categories to realize class-wise setting. For rotation transformation $\mathcal{R} \in \mathbb{R}^{3 \times 3}$, we refer to $\alpha$ and $\beta$ as slight angles imposed on the $x$ and $y$ axes, $\gamma$ as the primary angle for $z$ axis. We generate random angles for $\mathcal{A}_N$ times in three directions:

$$\alpha, \beta \sim \mathcal{U}(0, r_s), \gamma \sim \mathcal{U}(0, r_p), \mathcal{A}_N = \left\lceil \sqrt[3]{N} \right\rceil \tag{3}$$

where $\mathcal{U}$ denotes uniform distribution, $N$ denotes the number of categories, $r_s$ is a small range that controls $\alpha$ and $\beta$, while $r_p$ is a large range that controls $\gamma$. $\mathcal{A}_N$ is computed in such a way to ensure that the number of combinations of three angles is greater than or equal to $N$. Concretely, in the rotation operation, each of the three directions has $\mathcal{A}_N$ distinct angles, which means that the final rotation matrix has $\mathcal{A}_N^3$ possible combinations. To satisfy the class-wise setup, $\mathcal{A}_N^3$ must be at least $N$, requiring $\mathcal{A}_N$ to be no less than $\left\lceil \sqrt[3]{N} \right\rceil$. Finally, we randomly select $N$ combinations of angles for the allocation. The scaling transformation $\mathcal{S} \in \mathbb{R}^{3 \times 3}$ resizes the position of each point in the 3D point cloud sample by a certain scaling factor $\lambda$, which is sampled $N$ times from a uniform distribution $\mathcal{U}$:

$$\lambda \sim \mathcal{U}(b_l, b_u) \tag{4}$$

where $b_l$ and $b_u$ represent the lower bound and upper bound of the scaling factor, respectively. For shear $\mathcal{H} \in \mathbb{R}^{3 \times 3}$ defined in Eq. (13), twisting $\mathcal{W} \in \mathbb{R}^{3 \times 3}$ defined in Eq. (16), the process of generating parameters within ranges $(\omega_l, \omega_u)$ and $(h_l, h_u)$ is consistent to scaling. The range of these parameters ensures the visual effect of the sample.

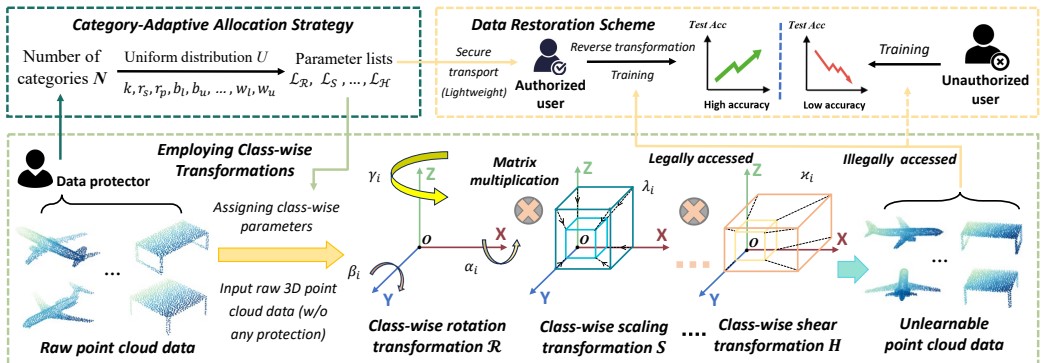

Figure 3: An overview of our proposed integral unlearnable pipeline

**Property 1.** *Since rotation matrices $\mathcal{R}_\alpha$, $\mathcal{R}_\beta$, and $\mathcal{R}_\gamma$ around three directions are all orthogonal matrices, $\mathcal{R}$ is also an orthogonal matrix, which can be defined as:*

$$\forall \mathcal{M} \in \{\mathcal{R}_\alpha, \mathcal{R}_\beta, \mathcal{R}_\gamma, \mathcal{R}\}, \quad \text{s.t.} \quad \mathcal{M}\mathcal{M}^T = \boldsymbol{I} \tag{5}$$

where we can determine the orthogonality by matrix multiplication through the definitions of Eq. (11). Since $\mathcal{R} = \mathcal{R}_\alpha \mathcal{R}_\beta \mathcal{R}_\gamma$ and $\mathcal{R}\mathcal{R}^T = \mathcal{R}_\alpha \mathcal{R}_\beta \mathcal{R}_\gamma \mathcal{R}_\gamma^T \mathcal{R}_\beta^T \mathcal{R}_\alpha^T = \boldsymbol{I}$, so $\mathcal{R}$ is also an orthogonal matrix.

**Property 2.** *All four transformation matrices we employ, $\mathcal{R}$, $\mathcal{S}$, $\mathcal{W}$, and $\mathcal{H}$, and the multiplicative combinations of any these matrices are all invertible matrices, which can be formally defined as:*

$$\forall \mathcal{J} \in \{f(\mathcal{R})f(\mathcal{S})f(\mathcal{W})f(\mathcal{H}) \mid f(x) \in \{x, \boldsymbol{I}\}\}, \quad \exists \mathcal{K} \quad \text{s.t.} \quad \mathcal{J}\mathcal{K} = \mathcal{K}\mathcal{J} = \boldsymbol{I} \tag{6}$$

where the inverse matrices of $\mathcal{R}$, $\mathcal{S}$, $\mathcal{W}$, and $\mathcal{H}$ are given in Appendix A. This property allows the authorized users to normally train on the protected data due to that multiplying a matrix by its inverse results in the identity matrix, leading us to propose a data restoration scheme in Sec. 3.4.

### 3.3.2 Employing Class-wise Transformations

Assuming the point cloud training set $\mathcal{D}_c$ is defined as $\{(\mathbf{X}_{c1}, \mathbf{Y}_i), (\mathbf{X}_{c2}, \mathbf{Y}_i), ..., (\mathbf{X}_{cn_i}, \mathbf{Y}_i)\}_{i=1}^N$, where $n_1, n_2, ..., n_N$ represent the number of samples in the 1st, 2nd, ..., $N$-th category, respectively. We formally define the spectrum of transformations as $k$ to indicate the number of transformations involved. Thus the ultimate unlearnable transformation matrix $\mathbf{T}_k$ is defined as:

$$\mathbf{T}_k = \prod_{i=1}^k \mathcal{V}_i, \quad \forall i \neq j, \quad \text{s.t.} \quad \mathcal{V}_i, \mathcal{V}_j \in \{\mathcal{R}, \mathcal{S}, \mathcal{W}, \mathcal{H}\}, \mathcal{V}_i \neq \mathcal{V}_j \tag{7}$$

Once we employ the proposed category-adaptive allocation strategy to $\mathbf{T}_k$, the unlearnable point cloud dataset $\mathcal{D}_u$ is constructed as:

$$\mathcal{D}_u = \{(\mathbf{T}_{k_i}(\mathbf{X}_{c1}), \mathbf{Y}_i), (\mathbf{T}_{k_i}(\mathbf{X}_{c2}), \mathbf{Y}_i), ..., (\mathbf{T}_{k_i}(\mathbf{X}_{cn_i}), \mathbf{Y}_i)\}_{i=1}^N \tag{8}$$

Our proposed UMT scheme is described in Algorithm 1. We enumerate possible transformations in Eq. (7) to obtain the unlearnability in Tab. 7 and select one type of class-wise transformation for each $k$ for more comprehensive experiments in Tab. 1. To facilitate the theoretical study of UMT[2], we opt for $\mathcal{R}\mathcal{S}$ as the transformation matrix $\mathbf{T}$, which achieves the best unlearnable effect as suggested in Tabs. 1 and 7. Thus, in the GMM scenario, the class-wise transformation matrix is defined as $\mathbf{T}_y = \mathcal{R}_y \mathcal{S}_y = \lambda_y \mathcal{R}_y \in \mathbb{R}^{d \times d}$, where $\lambda_y \in \mathbb{R}$ is the scaling factor.

**Lemma 3.** *The unlearnable dataset $\mathcal{D}_u$ generated using UMT on $\mathcal{D}_c$ can also be represented using a GMM, i.e., $\mathcal{D}_u \sim \mathcal{N}(y\mathbf{T}_y\mu, \lambda_y^2\boldsymbol{I})$.*

*Proof:* See Appendix D.1. Lemma 3 demonstrates that the unlearnable dataset $\mathcal{D}_u$ can also be represented as a GMM, which is derived from Property 1.

**Lemma 4.** *The Bayes optimal decision boundary for classifying $\mathcal{D}_u$ is given by $P_u(x) \equiv \mathcal{A}x^\top x + \mathcal{B}^\top x + \mathcal{C} = 0$, where $\mathcal{A} = \lambda_{-1}^{-2} - \lambda_1^{-2}$, $\mathcal{B} = 2(\lambda_{-1}^{-2}\mathbf{T}_{-1} + \lambda_1^{-2}\mathbf{T}_1)\mu$, and $\mathcal{C} = \ln\frac{|\lambda_{-1}^2\boldsymbol{I}|}{|\lambda_1^2\boldsymbol{I}|}$.*

---

[2]In the subsequent theoretical descriptions, UMT refers to UMT with $k=2$ using transformation matrix $\mathcal{R}\mathcal{S}$.

*Proof:* See Appendix D.2. Lemma 4 reveals that the Bayesian decision boundary for classifying $\mathcal{D}_u$ is a quadratic surface based on the GMM expression of $\mathcal{D}_u$.

**Lemma 5.** *Let $z \sim \mathcal{N}(0, \boldsymbol{I})$, $Z = z^\top z + b^\top z + c$, where $b = \frac{\mathcal{B}}{\mathcal{A}}, c = \frac{\mathcal{C}}{\mathcal{A}}$, and $\|\cdot\|_2$ denote 2-norm of vectors. For any $t \geq 0$ and $\gamma \in \mathbb{R}$, we employ Chernoff bound to have:*

$$\mathbb{P}\{Z \geq \mathbb{E}[Z] + \gamma\} \leq \frac{\exp\left\{\frac{t^2}{2(1-2t)}\|b\|_2^2 - t(\gamma + d)\right\}}{|(1-2t)\boldsymbol{I}|^{\frac{1}{2}}}$$

*Proof:* See Appendix D.3. Lemma 5 enables us to establish an upper bound on the accuracy of the unlearnable decision boundary $P_u$ applied to the clean dataset $\mathcal{D}_c$, denoted as $\tau_{\mathcal{D}_c}(P_u)$, as presented in Theorem 6 below.

**Theorem 6.** *For any constant $t_1$ and $t_2$ satisfying $0 \leq t_1 < \frac{1}{2}$ and $0 \leq t_2 < \frac{1}{2}$, the accuracy of the unlearnable decision boundary $P_u$ on $\mathcal{D}_c$ can be upper-bounded as:*

$$\tau_{\mathcal{D}_c}(P_u) \leq \frac{\exp\left\{\frac{t_1^2}{2(1-2t_1)}\|b + 2\mu\|_2^2 + t_1(\mu^\top \mu + b^\top \mu + c)\right\}}{2|(1-2t_1)\boldsymbol{I}|^{\frac{1}{2}}}$$

$$+ \frac{\exp\left\{\frac{t_2^2}{2(1-2t_2)}\|b - 2\mu\|_2^2 - t_2(\mu^\top \mu - b^\top \mu + c + 2d)\right\}}{2|(1-2t_2)\boldsymbol{I}|^{\frac{1}{2}}}$$

$$:= p_1 + p_2$$

*Furthermore, if $\mu^\top \mu + b^\top \mu + c + d < 0$ and $-\mu^\top \mu + b^\top \mu - c - d < 0$, we have $\tau_{\mathcal{D}_c}(P_u) < 1$. Moreover, for any $\mu \neq 0$, $\exists$ matrix $\mathbf{T}_i$ such that $\tau_{\mathcal{D}_c}(P_u) < \tau_{\mathcal{D}_c}(P)$, where $P$ is the Bayes optimal decision boundary for classifying $\mathcal{D}_c$.*

*Proof:* See Appendix D.4. The unlearnable effect takes place when $\tau_{\mathcal{D}_c}(P_u) < \tau_{\mathcal{D}_c}(P)$. To achieve this, we elaborately choose $\mathbf{T}_y$, which is formalized as $\mu^\top \frac{\lambda_{-1}^{-2}\mathbf{T}_{-1}^\top + \lambda_1^{-2}\mathbf{T}_1^\top}{\lambda_{-1}^{-2} - \lambda_1^{-2}}\mu \ll 0$. Therefore, Theorem 6 theoretically explains why UMT is effective in generating unlearnable point cloud data.

### 3.4 Data Restoration Scheme

To ensure that authorized users can achieve better generalization after training on unlearnable data, *i.e.*, satisfying Eq. (2), we exploit the inverse properties of 3D transformations, presented in Property 2, to calculate the inverse matrix of $\mathbf{T}_k$ as:

$$\mathbf{T}_k^{-1} = \prod_{i=k}^{1} \mathcal{V}_i^{-1}, \quad \forall i \neq j, \quad \text{s.t.} \quad \mathcal{V}_i^{-1}, \mathcal{V}_j^{-1} \in \{\mathcal{R}^{-1}, \mathcal{S}^{-1}, \mathcal{W}^{-1}, \mathcal{H}^{-1}\}, \mathcal{V}_i^{-1} \neq \mathcal{V}_j^{-1} \quad (9)$$

In particular, we note that $\mathcal{R}^{-1} = \mathcal{R}^T$, $\mathcal{S}^{-1} = \frac{1}{\lambda}\boldsymbol{I}$. Afterwards, the authorized user receives a lightweight message $M$ containing class-wise parameters from the data protector through a secure channel, thereby assigning $M$ to the inverse transformation matrix in Eq. (9) for multiplying the unlearnable samples. Our proposed integral unlearnable process is illustrated in Fig. 3.

## 4 Experiments

### 4.1 Experimental Details

**Datasets and Models.** Three synthetic 3D point cloud datasets, ModelNet40 [50], ModelNet10 [50], ShapeNetPart [4], and three real-world datasets including autonomous driving dataset KITTI [32] and indoor datasets ScanObjectNN [41], S3DIS [2] are used. We choose 16 widely used 3D point cloud models PointNet [35], PointNet++ [36], DGCNN [45], PointCNN [25], PCT [16], PointConv [48], CurveNet [51], SimpleView [15], 3DGCN [26], LGR-Net [59], RIConv [57], RIConv++ [58], PointMLP [31], PointNN [56], PointTransformerV3 [49], and SegNN [62] for evaluation of classification and semantic segmentation tasks.

**Experimental Setup.** The training process involves Adam optimizer [22], CosineAnnealingLR scheduler [30], initial learning rate of 0.001, weight decay of 0.0001. We empirically set $r_s$, $r_p$, $b_l$, $b_u$, $\omega_l$, $\omega_u$, $h_l$, and $h_u$ to 15°, 120°, 0.6, 0.8, 0°, 20°, 0, and 0.4 respectively. The main results of different

Table 1: **Main results:** The average test accuracy (%) results with standard deviations from three runs (random seeds are set to 23, 1023, and 2023) of classification models trained on the UMT datasets

| Datasets | Schemes | PointNet | PointNet++ | DGCNN | PointCNN | PCT | PointConv | CurveNet | SimpleView | RIConv++ | 3DGCN | PointNN | PointMLP | AVG |
|---|---|---|---|---|---|---|---|---|---|---|---|---|---|---|
| ModelNet10 | Clean | 89.85±0.54 | 92.11±1.57 | 91.81±0.84 | 87.99±1.58 | 90.18±2.23 | 91.26±1.12 | 91.88±1.51 | 89.52±0.41 | 87.90±0.64 | 88.51±5.39 | 83.22±0.73 | 91.07±0.19 | 89.61±0.42 |
| | UMT(k=1) | 40.64±10.69 | 28.73±2.00 | 27.41±6.71 | 32.82±2.70 | 31.58±4.27 | 33.64±6.63 | 39.78±6.80 | 45.31±11.85 | 86.72±1.46 | 28.01±8.72 | 34.47±0.29 | 32.85±2.35 | 38.50±2.01 |
| | UMT(k=2) | 21.18±0.93 | 26.36±1.71 | 18.84±6.14 | 21.97±4.04 | 19.72±4.52 | 20.84±5.96 | 25.04±2.15 | 22.75±2.43 | 16.53±3.63 | 32.79±3.23 | 31.94±2.11 | 25.41±3.96 | 23.61±1.06 |
| | UMT(k=3) | 22.84±1.71 | 23.81±8.43 | 29.16±4.30 | 27.03±9.97 | 29.43±9.11 | 18.49±7.39 | 25.51±9.57 | 32.12±12.58 | 19.94±1.13 | 36.19±13.38 | 30.54±1.24 | 32.21±5.28 | 27.27±6.20 |
| | UMT(k=4) | 18.83±2.40 | 20.56±14.61 | 15.92±1.51 | 20.52±6.06 | 20.29±2.14 | 21.66±2.24 | 23.46±12.58 | 24.12±6.20 | 26.41±2.47 | 34.28±17.20 | 25.59±0.25 | 26.65±11.36 | 23.19±5.98 |
| ModelNet40 | Clean | 86.18±0.07 | 90.55±0.73 | 89.51±0.86 | 81.89±5.35 | 87.11±1.39 | 88.90±0.89 | 87.82±0.23 | 85.25±0.31 | 84.59±1.07 | 86.81±1.69 | 74.81±0.16 | 87.31±0.91 | 85.89±0.40 |
| | UMT(k=1) | 28.62±1.80 | 20.69±0.87 | 28.60±1.65 | 21.92±1.86 | 29.06±0.38 | 24.99±2.63 | 33.29±3.92 | 26.89±1.48 | 75.86±8.43 | 26.30±1.92 | 28.57±0.39 | 29.28±1.72 | 31.17±0.20 |
| | UMT(k=2) | 8.30±0.87 | 18.71±1.65 | 11.60±1.97 | 10.85±2.61 | 9.21±2.31 | 12.99±0.99 | 12.71±3.70 | 12.05±3.88 | 10.48±2.01 | 26.17±0.35 | 25.05±0.06 | 10.90±4.74 | 14.09±0.78 |
| | UMT(k=3) | 7.48±1.66 | 17.62±3.10 | 10.41±5.48 | 9.43±3.09 | 9.91±4.19 | 11.28±4.93 | 11.23±3.07 | 11.21±4.09 | 10.92±0.48 | 25.31±1.29 | 24.53±0.10 | 8.94±3.43 | 13.19±2.41 |
| | UMT(k=4) | 7.83±2.25 | 15.72±4.12 | 10.86±2.57 | 8.60±2.28 | 9.68±1.58 | 10.52±2.09 | 10.91±3.00 | 14.32±0.92 | 18.09±3.83 | 17.16±5.47 | 24.31±0.28 | 12.38±1.51 | 13.36±0.43 |
| ShapeNetPart | Clean | 98.21±0.03 | 98.50±0.12 | 98.06±0.65 | 97.54±0.22 | 96.38±0.84 | 98.23±0.23 | 98.42±0.08 | 96.26±0.16 | 96.93±0.09 | 96.18±1.90 | 94.66±0.11 | 98.39±0.19 | 97.46±0.22 |
| | UMT(k=1) | 63.85±12.71 | 50.30±21.71 | 64.71±13.44 | 51.95±14.13 | 54.39±3.36 | 29.34±0.14 | 71.38±9.34 | 56.22±6.48 | 96.93±0.09 | 37.90±9.67 | 58.44±0.61 | 47.32±8.26 | 56.89±3.00 |
| | UMT(k=2) | 18.41±6.45 | 45.61±4.64 | 25.99±2.73 | 28.97±3.73 | 37.72±16.26 | 25.60±8.59 | 26.49±8.58 | 38.38±5.96 | 5.05±3.63 | 32.21±13.01 | 49.82±1.56 | 34.19±4.43 | 30.70±1.43 |
| | UMT(k=3) | 32.50±10.03 | 39.64±12.52 | 37.29±11.20 | 46.77±19.59 | 43.52±22.24 | 31.28±4.67 | 37.27±10.30 | 41.51±2.66 | 6.01±6.11 | 47.84±7.05 | 50.85±0.07 | 47.80±21.37 | 38.52±6.17 |
| | UMT(k=4) | 23.72±15.04 | 23.30±4.43 | 36.18±10.00 | 34.52±16.15 | 45.07±18.99 | 29.48±11.09 | 29.79±4.79 | 40.04±9.97 | 32.22±1.38 | 33.06±28.16 | 51.91±0.03 | 40.93±21.81 | 35.02±11.23 |
| KITTI | Clean | 98.04±2.23 | 99.49±0.50 | 99.33±0.34 | 99.23±0.25 | 98.93±0.54 | 98.04±1.68 | 99.10±1.05 | 99.38±0.33 | 99.64±0.09 | 99.67±0.24 | 99.49±0.50 | 95.72±5.46 | 98.84±0.87 |
| | UMT(k=1) | 36.23±30.18 | 62.90±20.51 | 38.93±33.08 | 57.80±13.99 | 58.26±21.66 | 39.99±6.82 | 33.33±10.75 | 56.71±30.96 | 98.53±1.19 | 68.34±10.46 | 70.34±0.08 | 47.20±20.79 | 55.71±5.54 |
| | UMT(k=2) | 31.24±7.11 | 51.07±20.29 | 27.90±10.33 | 49.69±11.52 | 51.47±17.34 | 25.95±11.38 | 20.55±4.93 | 51.77±8.53 | 99.80±0.09 | 70.42±27.97 | 47.31±10.58 | 39.90±10.00 | 47.26±5.95 |
| | UMT(k=3) | 19.13±6.93 | 53.73±12.44 | 31.26±21.68 | 56.08±17.90 | 54.45±1.77 | 42.54±21.46 | 27.38±19.20 | 32.62±11.76 | 98.48±1.61 | 69.51±8.27 | 70.14±1.30 | 58.44±6.61 | 51.15±4.79 |
| | UMT(k=4) | 26.84±16.26 | 61.79±6.65 | 29.89±15.72 | 54.27±11.99 | 64.21±10.76 | 57.29±12.05 | 37.70±11.60 | 54.63±11.07 | 99.34±0.23 | 72.33±3.47 | 70.49±0.76 | 56.25±12.33 | 57.09±5.97 |
| ScanObjectNN | Clean | 65.20±1.49 | 77.38±2.60 | 72.66±2.56 | 66.96±6.42 | 50.75±15.16 | 75.42±2.06 | 70.92±1.06 | 51.93±2.89 | 66.34±1.21 | 73.84±2.75 | 58.17±0.30 | 73.32±1.39 | 66.91±0.72 |
| | UMT(k=1) | 56.55±0.11 | 63.01±2.38 | 57.93±3.11 | 51.97±11.99 | 47.05±7.12 | 56.71±3.61 | 65.49±3.15 | 38.96±2.08 | 62.33±10.68 | 52.89±1.13 | 54.56±0.17 | 62.96±4.19 | 55.87±0.95 |
| | UMT(k=2) | 14.55±1.81 | 49.41±3.44 | 20.96±3.99 | 13.61±4.97 | 15.10±2.05 | 20.78±4.74 | 21.35±2.59 | 20.73±9.64 | 10.76±1.76 | 52.37±5.62 | 48.25±0.20 | 16.32±4.78 | 25.35±0.49 |
| | UMT(k=3) | 10.92±3.87 | 41.96±3.84 | 14.62±5.56 | 10.62±2.35 | 22.02±7.83 | 19.96±3.13 | 21.43±8.67 | 11.79±3.96 | 11.69±1.41 | 56.32±0.83 | 48.94±0.70 | 23.05±16.32 | 24.44±2.46 |
| | UMT(k=4) | 5.77±2.10 | 34.65±8.31 | 12.16±4.84 | 9.87±0.95 | 17.86±11.63 | 12.79±7.46 | 15.28±4.86 | 20.09±6.26 | 31.83±2.20 | 46.06±14.07 | 45.27±0.17 | 8.76±3.18 | 21.70±3.61 |

Table 2: **Robustness results:** The test accuracy (%) results on UMT-ModelNet40 against pre-process schemes

| Pre-process schemes | PointNet | PointNet++ | DGCNN | PointCNN | CurveNet | SimpleView | RIConv++ | 3DGCN | PointNN | PointMLP | AVG |
|---|---|---|---|---|---|---|---|---|---|---|---|
| Clean baseline | 86.10 | 91.13 | 89.02 | 75.73 | 87.76 | 85.49 | 85.82 | 84.89 | 75.00 | 87.66 | 84.86 |
| SOR | 9.93 | 19.17 | 9.36 | 10.45 | 11.57 | 14.81 | 6.66 | 25.08 | 25.04 | 15.22 | 14.73 |
| SRS | 9.24 | 22.20 | 10.25 | 7.78 | 12.66 | 14.04 | 11.32 | 24.72 | 26.01 | 12.18 | 15.04 |
| Random rotation | 10.94 | 26.74 | 11.59 | 10.21 | 12.13 | 6.86 | 12.09 | 55.64 | 29.74 | 20.37 | 19.63 |
| Random scaling | 22.33 | 22.45 | 30.15 | 20.22 | 25.61 | 23.25 | 73.62 | 27.48 | 26.50 | 24.39 | 29.60 |
| Random jitter | 9.64 | 21.72 | 10.05 | 7.74 | 9.98 | 16.68 | 13.27 | 23.74 | 26.90 | 9.09 | 14.88 |
| Random rotation & scaling | 63.49 | 34.44 | 44.65 | 37.56 | 72.89 | 51.62 | 78.04 | 64.00 | 36.26 | 78.94 | 56.19 |

$k$ on unlearnability is shown in Tab. 1. Specifically, $k = 1$ uses $\mathcal{R}$, $k = 2$ uses $\mathcal{RS}$, $k = 3$ uses $\mathcal{RSW}$, and $k = 4$ uses $\mathcal{RSWH}$. More results for different combinations of class-wise transformations are provided in Tab. 7. The table values covered by gray denote the best unlearnable effect.

**Evaluation Metrics.** For classification, we report the *test accuracy* (in %) derived from the classification accuracy on clean test set, which aligns with the metrics used in the 2D unlearnable schemes [19, 28, 39]. For semantic segmentation, we use *eval accuracy* and *mean Intersection over Union* (mIoU) as evaluation metrics (in %), where *eval accuracy* is the ratio of the number of points classified correctly to the total number of points in the point cloud, mIoU calculates the IoU for each class between the ground truth and the predicted segmentation [61], and then takes the average of these ratios. The lower these metrics are, the better the effect of the unlearnable scheme.

### 4.2 Evaluation of Proposed Unlearnable Schemes

**Effectiveness.** As shown in Tab. 1, UMT results in a significant decrease in test accuracy compared to clean baseline, indicating the unlearnable effectiveness of UMT. Moreover, all values of $k$ achieve good unlearnable results. The average performance of $\mathcal{RS}$ is the best, which is employed as the default in the remaining experiments. We note that rigid transformations are easily defeated by invariance networks [10, 26, 58]. Therefore, in practical settings, we will include non-rigid transformations, using combined transformations to enhance the robustness of the UMT scheme.

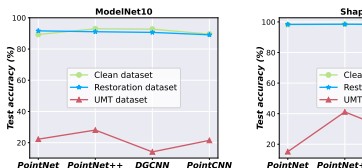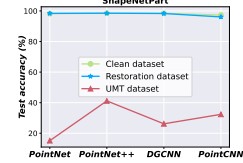

Figure 4: The test accuracy (%) results obtained after training on the clean, UMT, and restoration datasets

**Robustness.** (i) *Data augmentations.* Similar to [18, 24], we employ data augmentations like *random scaling*, *random jitter*, *random rotation*, *Statistic Outlier Removal* (SOR) [60], and *Simple Random Sampling* (SRS) [53] against UMT. Tab. 2 suggests UMT is robust to random data augmentations. SOR detects and removes outliers or noisy points but is ineffective in countering UMT because UMT does not introduce irregular perturbations or add outliers. SRS randomly selects a small subset from the entire set of points with equal probability. UMT is robust to SRS as it alters the coordinates of all points. Even if an arbitrary subset is chosen, all points within the subset have already undergone the unlearnable transformations. (ii) *Adaptive attack.* We assume the unauthorized user gains

Table 3: **Semantic segmentation:** Evaluation of UMT on semantic segmentation task using S3DIS dataset

| Evaluation metrics | Eval accuracy (%) | | | | mIoU (%) | | | |
|---|---|---|---|---|---|---|---|---|
| Segmentation models | PointNet++ [36] | Point Transformer v3 [49] | SegNN [62] | AVG | PointNet++ [36] | Point Transformer v3 [49] | SegNN [62] | AVG |
| Clean baseline | 74.76 | 74.72 | 79.00 | 76.16 | 40.06 | 40.57 | 50.27 | 43.63 |
| **UMT**$_{(k=2)}$ | **15.89** | **25.61** | **66.41** | **35.97** | **7.45** | **18.32** | **46.30** | **24.02** |

Table 4: **Ablation study:** The test accuracy (%) results derived from unlearnable data with different modules

| Datasets | Modules↓ Models→ | PointNet | PointNet++ | DGCNN | PointCNN | CurveNet | SimpleView | RIConv | LGR-Net | 3DGCN | PointNN | PointMLP | AVG |
|---|---|---|---|---|---|---|---|---|---|---|---|---|---|
| ModelNet40 | Module $\mathcal{R}$ | 27.19 | 20.02 | 28.61 | 21.64 | 36.87 | 28.32 | 88.25 | 40.24 | 24.59 | 28.77 | 27.72 | 33.84 |
| | Module $\mathcal{S}$ | 9.36 | 48.70 | 15.07 | 7.41 | 13.92 | 20.33 | 32.25 | 12.56 | 88.72 | 65.32 | 18.79 | 30.22 |
| | UMT$_{(k=2)}$ | **9.20** | **18.52** | **13.86** | 9.08 | **13.90** | 7.58 | 24.39 | **9.85** | 26.58 | 25.12 | 7.31 | **15.04** |
| ModelNet10 | Module $\mathcal{R}$ | 29.85 | 30.51 | 35.13 | 31.17 | 47.25 | 58.70 | 91.38 | 43.72 | 36.50 | 34.58 | 30.47 | 42.66 |
| | Module $\mathcal{S}$ | 34.80 | 57.05 | 42.62 | 33.81 | 30.36 | 43.75 | 42.24 | 32.60 | 89.40 | 73.79 | 41.29 | 47.43 |
| | UMT$_{(k=2)}$ | **22.25** | **28.08** | **14.10** | **21.48** | **24.34** | **21.37** | **38.18** | **19.82** | **31.28** | **33.37** | **29.35** | **25.78** |

knowledge about the $\mathcal{G}_p$'s use of $\mathcal{RS}$. Thus we propose random rotation & scaling as an adaptive scheme. In Tab. 2, the adaptive scheme exhibits a higher accuracy than other schemes, confirming its effectiveness. Nonetheless, it remains 28.67% lower than clean baseline, revealing the robustness of UMT against adaptive attack. More results of adaptive attacks are provided in Appendix C.3.

**Visual Effect.** We visualize UMT samples in Figs. 7 to 10, indicating that the unlearnable point cloud samples still retain their normal feature structure with visual rationality.

**Evaluation of Semantic Segmentation.** We evaluate UMT using common metrics for point cloud semantic segmentation tasks in Tab. 3. As can be seen, the performance of semantic segmentation of data protected by UMT significantly decreases. The underlying reason is that the DNNs learn the features of class-wise transformations and establish a new mapping, which leads to the inability of test samples without transformations to be correctly segmented by the segmentation model.

**Evaluation of Data Restoration.** We multiply UMT samples by the transformation matrix in Eq. (9). The data becomes learnable after the restoration process, with test accuracy reaching a level comparable to the clean baseline as shown in Fig. 4. This strongly validates the effectiveness of the proposed data restoration scheme.

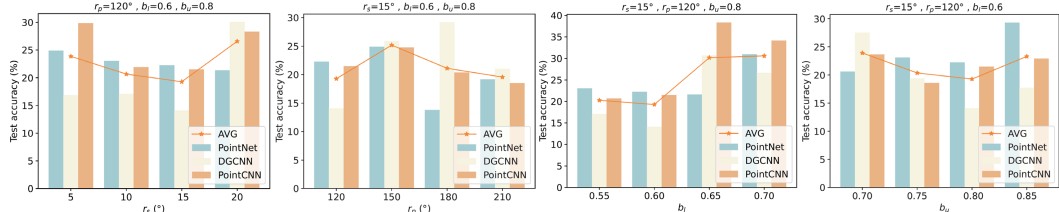

Figure 5: **Hyper-parameter sensitivity analysis:** The impact of hyperparameters $r_s$, $r_p$, $b_l$, and $b_u$ on the test accuracy results (%) on the UMT (using $\mathcal{RS}$) ModelNet10 dataset

## 4.3 Ablation Study and Hyper-Parameter Sensitivity Analysis

**Ablation on Rotation Module.** As shown in Tab. 4, the average accuracy increases by 15.18% and 21.65%, respectively, when only using $\mathcal{S}$. This suggests the importance of class-wise rotation module. The high test accuracy demonstrated by 3DGCN [26] can be attributed to its scaling invariance, which endows it with robustness against scaling transformation.

**Ablation on Scaling Module.** As also shown in Tab. 4, the average accuracy increases by 18.80% and 16.88% when only using the rotation module, respectively. The high test accuracy achieved on RIConv [57] and LGR-Net [59] is due to the fact that both networks are rotation-invariant, thus providing resistance against rotation transformations. These ablation results furthermore emphasize the importance of incorporating more non-rigid transformations.

**Hyper-Parameter Analysis.** We analyze four hyperparameters $r_s$, $r_p$, $b_l$, and $b_u$ in Fig. 5. The influence of $r_s$ and $r_p$ on the accuracy remains relatively small, exhibiting their best unlearnable effect when set to 15° and 120°, respectively. We attribute this to the crucial role played by the class-wise setting, while it is not highly sensitive to the size of specific values. The unlearnable effect is the best when $b_l$ and $b_u$ are set to 0.6 and 0.8, respectively. Similarly, the variations in $b_l$ and $b_u$ do not significantly alter the effect due to the class-wise setting.

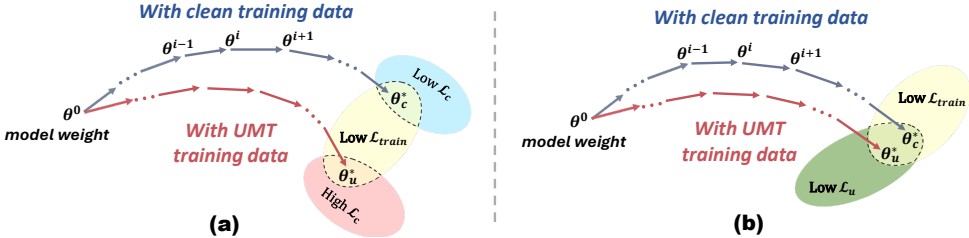

Figure 6: UMT in weight space. The blue arrow represents the clean training trajectory of the weights $\theta_i$ at step $i$, while the red arrows denote the UMT training trajectory. The values for plotting this figure are provided in Appendix C.6. (a) Testing on clean test set (*blue and red ellipses*); (b) Testing on UMT test set (*green ellipse*)

## 4.4 Insightful Analysis Into UMT

We formalize $\mathcal{L}(\mathcal{D}) = \mathbb{E}_{(\mathbf{X},\mathbf{Y})\sim\mathcal{D}}[\mathcal{L}_c(\mathcal{F}(\mathbf{X};\theta), \mathbf{Y})]$, where $\mathcal{L}_c$ is the cross-entropy loss, $\mathbf{X}, \mathbf{Y}$ is the point cloud data sampled from dataset $\mathcal{D}$. We define $\mathcal{D}_{tr}$, $\mathcal{D}_u$, and $\mathcal{D}_c$ as the training set, unlearnable test set (*i.e.*, test set transformed by UMT), and clean test set, respectively. Thus we have the *training loss $\mathcal{L}_{train} = \mathcal{L}(\mathcal{D}_{tr})$, unlearnable test loss $\mathcal{L}_u = \mathcal{L}(\mathcal{D}_u)$, and clean test loss $\mathcal{L}_c = \mathcal{L}(\mathcal{D}_c)$.*

The models trained on both clean and UMT training sets exhibit low $\mathcal{L}_{train}$ as shown in Fig. 6 (*yellow ellipses*), indicating the models converge well during training. Furthermore, when tested on $\mathcal{D}_c$ as shown in Fig. 6 (a), the clean model (trained with clean training set) achieves a low $\mathcal{L}_c$ (*blue ellipse*), while the UMT model (trained with UMT training set) exhibits a high $\mathcal{L}_c$ (*red ellipse*), also supporting the unlearnable effectiveness of UMT. Fig. 6 (b) reveals that the clean model and UMT model both exhibit low $\mathcal{L}_u$ (*green ellipse*), suggesting that they can classify UMT samples. But the mechanisms underlying the two cases of low $\mathcal{L}_u$ differ. The clean one is due to that the semantics of samples can be remained by UMT and thus normally classified. The UMT one is that the UMT model learns the mapping between transformations and labels, thereby correctly predicting the samples containing the same transformations. We conclude that the clean model effectively classifies both clean and UMT samples, while the UMT model successfully classifies UMT samples (the UMT process is the same for both training and test samples) but fails to classify clean samples.

## 5 Related Work

### 5.1 2D Unlearnable Schemes

The development of 2D unlearnable schemes [19, 29, 37, 39, 42, 55] has been booming. Specifically, model-dependent methods are initially proposed in abundance [9, 19, 29]. Afterwards, numerous model-agnostic methods that significantly improve the generation efficiency have surfaced [39, 42, 47]. However, due to the structural disparities between 3D point cloud data and 2D images, applying unlearnable methods directly from 2D to 3D reveals significant challenges.

### 5.2 Protecting 3D Point Cloud Data Privacy

Some works proposed an encryption scheme based on chaotic mapping [21] or optical chaotic encryption [27], and a 3D object reconstruction technique was introduced [34], both achieving privacy protection for individual 3D point cloud data. Nevertheless, no privacy-preserving solution has been proposed specifically for the scenario of unauthorized DNN learning on abundant raw 3D point cloud data. It is worth mentioning that both parallel works [44, 63] study availability poisoning attacks against 3D point cloud networks, which largely reduce model accuracy, and both have the potential to be applied as unlearnable schemes. However, the feature collision error-minimization poisoning scheme proposed by Zhu *et al.* [63] overlooks the problem of effective training for authorized users, which limits its practical use in real-world applications. The rotation-based poisoning approach proposed by Wang *et al.* [44] is easily defeated by rotation-invariant networks [57, 58, 59], as revealed in Tabs. 1 and 4.

# 6 Conclusion, Limitation, and Broader Impacts

In this research, we propose the first integral unlearnable framework in 3D point clouds, which utilizes class-wise multi transformations, preventing unauthorized deep learning while allowing authorized training. Extensive experiments on synthetic and real-world datasets and theoretical evidence verify the superiority of our framework. The transformations include rotation, scaling, twisting, and shear, which are all common 3D transformation operations. If unauthorized users design a network that is invariant to all these transformations, they could potentially defeat our proposed UMT. So far, only networks invariant to rigid transformations like rotation and scaling have been proposed, while networks invariant to non-rigid transformations like twisting and shear, have not yet been introduced. Therefore, our research also contributes to the design of more transformation-invariant networks.

Our research calls for the design of more robust point cloud networks, which helps improve the robustness and security of 3D point cloud processing systems. On the other hand, if our proposed UMT scheme is maliciously exploited, it may have negative impacts on society, such as causing a sharp decline in the performance of models trained on it, affecting the security and reliability of technologies based on 3D point cloud networks. More transformation-invariant 3D point cloud networks need to be proposed in the future to avoid potential negative impacts.

## Acknowledgements

Shengshan's work is supported by the National Natural Science Foundation of China (Grant Nos. U20A20177, 62372196). Minghui Li is the corresponding author.

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

# Appendix: Unlearnable 3D Point Clouds: Class-wise Transformation Is All You Need

## A Definitions of 3D Transformations

Existing 3D transformations are summarized in Fig. 1 and formally defined in this section. The 3D transformations are mathematical operations applied to three-dimensional objects to change their position, orientation, and scale in space, which are often represented using transformation matrices. We formally define the transformed point cloud sample with transformation matrix $\mathbf{T} \in \mathbb{R}^{3 \times 3}$ as:

$$\mathbf{X}_t = \mathbf{T} \cdot \mathbf{X} \tag{10}$$

where $\mathbf{X} \in \mathbb{R}^{3 \times p}$ is the clean point cloud sample, $\mathbf{X}_t \in \mathbb{R}^{3 \times p}$ is the transformed point cloud sample.

### A.1 Rotation Transformation

The rotation transformation that alters the orientation and angle of 3D point clouds is controlled by three angles $\alpha$, $\beta$, and $\gamma$. The rotation matrices in three directions can be formally defined as:

$$\mathcal{R}_\alpha = \begin{bmatrix} 1 & 0 & 0 \\ 0 & \cos\alpha & -\sin\alpha \\ 0 & \sin\alpha & \cos\alpha \end{bmatrix}, \mathcal{R}_\beta = \begin{bmatrix} \cos\beta & 0 & \sin\beta \\ 0 & 1 & 0 \\ -\sin\beta & 0 & \cos\beta \end{bmatrix}, \mathcal{R}_\gamma = \begin{bmatrix} \cos\gamma & -\sin\gamma & 0 \\ \sin\gamma & \cos\gamma & 0 \\ 0 & 0 & 1 \end{bmatrix} \tag{11}$$

Thus we have $\mathbf{T} = \mathcal{R}_\alpha \mathcal{R}_\beta \mathcal{R}_\gamma$ while employing rotation transformation. Besides, we have $\mathcal{R}_\alpha^{-1} = \mathcal{R}_\alpha^T$, $\mathcal{R}_\beta^{-1} = \mathcal{R}_\beta^T$, and $\mathcal{R}_\gamma^{-1} = \mathcal{R}_\gamma^T$.

### A.2 Scaling Transformation

The scaling matrix $\mathcal{S}$ can be represented as:

$$\mathcal{S} = \begin{bmatrix} \lambda & 0 & 0 \\ 0 & \lambda & 0 \\ 0 & 0 & \lambda \end{bmatrix} = \lambda \begin{bmatrix} 1 & 0 & 0 \\ 0 & 1 & 0 \\ 0 & 0 & 1 \end{bmatrix} \tag{12}$$

where the scaling factor $\lambda$ is used to perform a proportional scaling of the coordinates of each point in the point cloud.

### A.3 Shear Transformation

In the three-dimensional space, shearing [3] is represented by different matrices, and the specific form depends on the type of shear being performed. Specifically, the shear transformation matrix $\mathcal{H}_{xy}$ (employed in UMT) of shifting $x$ and $y$ by the other coordinate $z$, and its corresponding inverse matrix can be expressed as:

$$\mathcal{H}_{xy} = \begin{bmatrix} 1 & 0 & s \\ 0 & 1 & t \\ 0 & 0 & 1 \end{bmatrix}, \mathcal{H}_{xy}^{-1} = \begin{bmatrix} 1 & 0 & -s \\ 0 & 1 & -t \\ 0 & 0 & 1 \end{bmatrix} \tag{13}$$

The shear transformation matrix $\mathcal{H}_{xz}$ of shifting $x$ and $z$ by the other coordinate $y$, and its corresponding inverse matrix can be expressed as:

$$\mathcal{H}_{xz} = \begin{bmatrix} 1 & s & 0 \\ 0 & 1 & 0 \\ 0 & t & 1 \end{bmatrix}, \mathcal{H}_{xz}^{-1} = \begin{bmatrix} 1 & -s & 0 \\ 0 & 1 & 0 \\ 0 & -t & 1 \end{bmatrix} \tag{14}$$

The shear transformation matrix $\mathcal{H}_{yz}$ of shifting $y$ and $z$ by the other coordinate $x$, and its corresponding inverse matrix can be expressed as:

$$\mathcal{H}_{yz} = \begin{bmatrix} 1 & 0 & 0 \\ s & 1 & 0 \\ t & 0 & 1 \end{bmatrix}, \mathcal{H}_{yz}^{-1} = \begin{bmatrix} 1 & 0 & 0 \\ -s & 1 & 0 \\ -t & 0 & 1 \end{bmatrix} \tag{15}$$

---

[3] https://www.mauriciopoppe.com/notes/computer-graphics/transformation-matrices/shearing/

**Algorithm 1** Our proposed UMT scheme

**Input:** Clean 3D point cloud dataset $\mathcal{D}_c = \{(\mathbf{X}_i, \mathbf{Y}_i)\}_{i=1}^{n}$; number of categories $N$; slight range $r_s$; primary range $r_p$; scaling lower bound $b_l$; scaling upper bound $b_u$; twisting lower bound $w_l$; twisting upper bound $w_u$; shear lower bound $h_l$; shear upper bound $h_u$; spectrum of transformations $k$; matrix set $\mathcal{T}_s = \{\mathcal{R}, \mathcal{S}, \mathcal{H}, \mathcal{W}\}$.
**Output:** Unlearnable 3D point cloud dataset $\mathcal{D}_u = \{(\mathbf{X}_{\mathbf{u}i}, \mathbf{Y}_i)\}_{i=1}^{n}$.
Initialize the slight angle lists $\mathcal{L}_\alpha$=[ ], $\mathcal{L}_\beta$=[ ], the primary angle list $\mathcal{L}_\gamma$=[ ], the rotation list $\mathcal{L}_\mathcal{R}$=[ ], the scaling list $\mathcal{L}_\mathcal{S}$=[ ], the twisting list $\mathcal{L}_\mathcal{W}$=[ ], the shear list $\mathcal{L}_\mathcal{H}$=[ ], and $\mathcal{A}_N = \left\lceil \sqrt[3]{N} \right\rceil$;

**for** $c = 1$ *to* $k$ **do**
    Randomly sample a transformation matrix $\mathcal{V}_c \in \mathcal{T}_s$;
    Remove transformation matrix $\mathcal{V}_c$ from $\mathcal{T}_s$;
    **if** $\mathcal{V}_c == \mathcal{R}$ **then**
        **for** $i = 1$ *to* $\mathcal{A}_N$ **do**
            **for** $j = 1$ *to* $\mathcal{A}_N$ **do**
                **for** $k = 1$ *to* $\mathcal{A}_N$ **do**
                    $\mathcal{L}_\mathcal{R} \leftarrow \mathcal{L}_\mathcal{R} \cup \{[\mathcal{L}_\alpha[i], \mathcal{L}_\beta[j], \mathcal{L}_\gamma[k]]\}$;
                **end**
            **end**
        **end**
        Get the $\mathcal{L}_\mathcal{R} \leftarrow random.sample(\mathcal{L}_\mathcal{R}, N)$;
    **end**
    **else if** $\mathcal{V}_c == \mathcal{S}$ **then**
        **for** $i = 1$ *to* $N$ **do**
            Randomly sample $\lambda_i \sim \mathcal{U}(b_l, b_u)$;
            Add to the list $\mathcal{L}_\mathcal{S} \leftarrow \mathcal{L}_\mathcal{S} \cup \{\lambda_i\}$;
        **end**
    **end**
    **else if** $\mathcal{V}_c == \mathcal{W}$ **then**
        **for** $i = 1$ *to* $N$ **do**
            Randomly sample $\omega_i \sim \mathcal{U}(w_l, w_u)$;
            Add to the list $\mathcal{L}_\mathcal{W} \leftarrow \mathcal{L}_\mathcal{W} \cup \{\omega_i\}$;
        **end**
    **end**
    **else if** $\mathcal{V}_c == \mathcal{H}$ **then**
        **for** $i = 1$ *to* $N$ **do**
            Randomly sample $h_i \sim \mathcal{U}(h_l, h_u)$;
            Add to the list $\mathcal{L}_\mathcal{H} \leftarrow \mathcal{L}_\mathcal{H} \cup \{h_i\}$;
        **end**
    **end**
**end**
**for** $i = 1$ *to* $n$ **do**
    Get the transformation matrix $\mathbf{T}_k = \prod_{i=1}^{k} \mathcal{V}_i$ by the parameter lists above;
    Get the transformed data $\mathbf{X}_{\mathbf{u}i} = \mathbf{T}_{ki} \cdot \mathbf{X}_i$;
**end**
**Return:** Unlearnable 3D point cloud dataset $\mathcal{D}_u$.

---

### A.4 Twisting Transformation

The 3D twisting transformation [7] involves a rotational deformation applied to an object in three-dimensional space, creating a twisted or spiraled effect. Unlike simple rotations around fixed axes, a twisting transformation introduces a variable rotation that may change based on the spatial coordinates of the object. For instance, considering a twisting transformation along the $z$-axis, where the rotation angle is a function related to the $z$-coordinate, it can be expressed as:

$$\mathcal{W}_z(\theta, z) = \begin{bmatrix} \cos(\theta z) & -\sin(\theta z) & 0 \\ \sin(\theta z) & \cos(\theta z) & 0 \\ 0 & 0 & 1 \end{bmatrix} \tag{16}$$

where $\theta$ is the parameter of the twisting transformation, and $z$ is the $z$-coordinate of the object. The inverse matrix of $\mathcal{W}_z(\theta, z)$ is:

$$\mathcal{W}_z^{-1}(\theta, z) = \begin{bmatrix} \cos(\theta z) & \sin(\theta z) & 0 \\ -\sin(\theta z) & \cos(\theta z) & 0 \\ 0 & 0 & 1 \end{bmatrix} \tag{17}$$

### A.5 Tapering Transformation

The tapering transformation [7] is a linear transformation used to alter the shape of an object, causing it to gradually become pointed or shortened. In three-dimensional space, tapering transformation can adjust the dimensions of an object along one or more axes, creating a tapering effect. The specific matrix representation of tapering transformation depends on the chosen axis and the design of the transformation. Generally, tapering transformation can be represented by a matrix that is multiplied by the coordinates of the object to achieve the shape adjustment, which is defined as:

$$\mathcal{A}_z(\eta, z) = \begin{bmatrix} 1 + \eta z & 0 & 0 \\ 0 & 1 + \eta z & 0 \\ 0 & 0 & 1 \end{bmatrix} \tag{18}$$

where $z$ is the $z$-coordinate of the object. Considering that $\eta z$ could indeed equal to -1, in such a case, the 3D point cloud samples would be projected onto the $z$-plane, losing their practical significance and the tapering matrix is also irreversible.

### A.6 Reflection Transformation

Reflection transformation is a linear transformation that inverts an object along a certain plane. This plane is commonly referred to as a reflection plane or mirror. For reflection transformations in three-dimensional space, we can represent them through a matrix. Regarding the reflection transformation matrices of the $xy$ plane, $yz$ plane, and $xz$ plane, we have:

$$\mathcal{R}_{xy} = \begin{bmatrix} 1 & 0 & 0 \\ 0 & 1 & 0 \\ 0 & 0 & -1 \end{bmatrix}, \mathcal{R}_{yz} = \begin{bmatrix} -1 & 0 & 0 \\ 0 & 1 & 0 \\ 0 & 0 & 1 \end{bmatrix}, \mathcal{R}_{xz} = \begin{bmatrix} 1 & 0 & 0 \\ 0 & -1 & 0 \\ 0 & 0 & 1 \end{bmatrix} \tag{19}$$

### A.7 Translation Transformation

The 3D translation transformation[4] refers to the process of moving an object in three-dimensional space. This transformation involves moving the object along the $x$, $y$, and $z$ axes, respectively, smoothly transitioning it from one position to another, which is defined as:

$$\mathcal{L} = \begin{bmatrix} 1 & 0 & 0 & t_x \\ 0 & 1 & 0 & t_y \\ 0 & 0 & 1 & t_z \\ 0 & 0 & 0 & 1 \end{bmatrix} \tag{20}$$

where $t_x$, $t_y$, and $t_z$ represents the translation along the $x$, $y$, and $z$ axes, respectively. This 4×4 matrix is a homogeneous coordinate matrix that describes the translation in three-dimensional space. Additionally, the translation matrix also can be represented as a additive matrix to original point $x_i \in \mathbb{R}^{3 \times 3}$, which can be defined as:

$$\mathcal{L} = \begin{bmatrix} t_x & t_x & t_x \\ t_y & t_y & t_y \\ t_z & t_z & t_z \end{bmatrix} \tag{21}$$

## B Supplementary Experimental Settings

### B.1 Experimental Platform

Our experiments are conducted on a server running a 64-bit Ubuntu 20.04.1 system with an Intel Xeon Silver 4210R CPU @ 2.40GHz processor, 125GB memory, and four Nvidia GeForce RTX 3090 GPUs, each with 24GB memory. The experiments are performed using the Python language, version 3.8.19, and PyTorch library version 1.12.1.

---

[4]https://www.javatpoint.com/computer-graphics-3d-transformations

Table 5: The test accuracy (%) results on diverse types of transformations on ModelNet10 training set using PointNet classifier

| Transformations | Rotation | Scaling | Shear | Twisting | Tapering | Translation | AVG |
|---|---|---|---|---|---|---|---|
| w/o | 89.32 | 89.32 | 89.32 | 89.32 | 89.32 | 89.32 | 89.32 |
| Sample-wise | 91.52 | 87.78 | 89.10 | 92.62 | 90.31 | 90.31 | 89.80 |
| Dataset-wise | 87.89 | 79.41 | 85.79 | 92.18 | 88.22 | 88.22 | 83.45 |
| Class-wise | 29.85 | 20.81 | 67.84 | 63.22 | 52.42 | 36.67 | 45.14 |

Table 6: Accuracy results obtained with different test sets when training the PointNet classifier using class-wise transformed training sets

| Test sets ↓ Transformations ⟶ | Rotation | Scaling | Shear | Twisting | Tapering | Translation |
|---|---|---|---|---|---|---|
| Class-wise test set | 99.67 | 93.80 | 97.91 | 94.05 | 96.04 | 98.24 |
| Permuted class-wise test set | 10.68 | 38.22 | 58.37 | 60.68 | 42.51 | 31.94 |
| Clean test set | 29.85 | 20.81 | 67.84 | 63.22 | 52.42 | 36.67 |

## B.2 Hyper-Parameter Settings

The model training process on the unlearnable dataset and the clean dataset remains consistent, using the Adam optimizer [22], CosineAnnealingLR scheduler [30], initial learning rate of 0.001, weight decay of 0.0001, batch size of 16 (due to insufficient GPU memory, the batch size is set to 8 when training 3DGCN on ModelNet40 dataset), and training for 80 epochs. Due to the longer training process required by PCT [16], the training epochs for PCT in Tab. 1 on ModelNet10, ModelNet40, and ScanObjectNN datasets are all set to 240.

## B.3 Settings of Exploring Experiments

We initially investigate which approach yields the best unlearnable effect among *sample-wise*, *dataset-wise*, and *class-wise* settings in Fig. 2 (a). Specific experimental settings are as follows:

In the *dataset-wise* setting, the same parameter values are applied to the entire dataset. Specifically, we have $\alpha = \beta = \gamma = 10°$ in the rotation transformation, the scaling factor $\lambda$ is set to 0.8, both shearing factors $s$ and $t$ are set to 0.2. The angle $\theta$ in twisting is $25°$. The tapering angle $\eta$ is set to $25°$, and the parameters in translation transformation $t_x$, $t_y$, and $t_z$ are set to 0.15.

In the *sample-wise* setting, each sample has its independent set of parameters, meaning the parameter values for each sample are randomly generated within a certain range. In the rotation transformation, $\alpha$, $\beta$, $\gamma$ are uniformly sampled from the range of $0°$ to $20°$. The scaling factor $\lambda$ is uniformly sampled from 0.6 to 0.8, shearing factors $s$ and $t$ are uniformly sampled from the range of 0 to 0.4. Both the twist angle $\theta$ and tapering angle $\eta$ are sampled from $0°$ to $50°$, and the parameters in translation transformation $t_x$, $t_y$, and $t_z$ are sampled from 0 to 0.3.

In the *class-wise* setting, the parameters for transformations are associated with the point cloud's class. The selection of parameters for each class is also obtained by random sampling within a fixed range. The chosen ranges are generally consistent with those described for the *sample-wise* setting above. However, a difference lies in the consideration of slight angle range and primary angle range in the case of rotation transformations, where these ranges are $20°$ and $120°$, respectively.

The specific results of test accuracy under different transformation modes, including sample-wise (random), class-wise, and dataset-wise (universal), are provided in Tab. 5. It can be seen that under the class-wise setting, the final unlearnable effect is the best.

## B.4 Benchmark Datasets

**Dataset Introduction.** The ModelNet40 dataset is a point cloud dataset containing 40 categories, comprising 9843 training and 2468 test point cloud data. ModelNet10 is a subset of ModelNet40 dataset with 10 categories. ShapeNetPart that includes 16 categories is a subset of ShapeNet, comprising 12137 training and 2874 test point cloud samples. ScanObjectNN is a real-world point cloud dataset with 15 categories, comprising 2309 training samples and 581 test samples. Similar to [17, 52], we split KITTI object clouds into class "vehicle" and "human" containing 1000 training

data and 662 test data. The input point cloud objects for the models encompass 256 points for the KITTI dataset and 1024 points for other datasets. The *Stanford 3D Indoor Scene Dataset* (S3DIS) dataset contains 6 large-scale indoor areas with 271 rooms. Each point in the scene point cloud is annotated with one of the 13 semantic categories.

**Dataset Licenses.** The dataset license information is listed as:

- **ModelNet10, ModelNet40 [50]:** All CAD models are downloaded from the Internet and the original authors hold the copyright of the CAD models. The label of the data is obtained by us via Amazon Mechanical Turk service and it is provided freely. This dataset is provided for the convenience of academic research only. Link is `https://modelnet.cs.princeton.edu/`.

- **ShapeNetPart [4]:** We use the ShapeNet database (the "Database") at Princeton University and Stanford University. Link is `https://www.shapenet.org/`.

- **ScanObjectNN [41]:** The license is MIT License: Copyright (c) 2019 Vision & Graphics Group, HKUST. Permission is hereby granted, free of charge, to any person obtaining a copy of this software and associated documentation files (the "Software"), to deal in the Software without restriction, including without limitation the rights to use, copy, modify, merge, publish, distribute, sublicense, and/or sell copies of the Software, and to permit persons to whom the Software is furnished to do so, subject to the following conditions: The above copyright notice and this permission notice shall be included in all copies or substantial portions of the Software. Link is `https://hkust-vgd.github.io/scanobjectnn/`.

- **S3DIS [2]:** The copyright is from Stanford University, Patent Pending, copyright 2016. Link is `http://buildingparser.stanford.edu/dataset.html`.

### B.5 Settings of Robustness Experiments

The scaling factor in random scaling augmentation is set to a minimum of 0.8 and a maximum of 1.25. In the random rotation operation, the three directional rotation angles are identical and uniformly sampled from $[0, 2\pi)$. The perturbations in the random jitter are sampled from a normal distribution with a standard deviation of 0.05, and the perturbation magnitude is constrained within 0.1. The parameters $k$ and $\alpha$ in SOR are set to 2 and 1.1, respectively. The number of dropped points in SRS is 500.

## C   Supplementary Experimental Results

### C.1 Results for Execution Mechanism

We present the specific experimental results of Fig. 2 (a) in Tab. 5. It can be observed that under the class-wise setting, the average test accuracy is the lowest, while the unlearnable condition yields the best performance. Furthermore, we conduct investigations into training the PointNet classifier with class-wise transformed training sets (ModelNet10 dataset), analyzing accuracy results across different test sets, as shown in Tab. 6. It is noteworthy that the accuracy on class-wise test sets (using consistent transformation parameters with class-wise transformed training set) is consistently above 90%, indicating that the model has learned the mapping between transformations and labels. This leads to correct classification on test samples with the same transformations. However, when the class-wise test set undergoes permutation, the accuracy drops to a level similar to that of the clean test set. This clearly demonstrates that the model can correctly classify samples only when they have corresponding transformations, and samples without transformations or with mismatched transformations cannot be correctly classified. This further validates that the model learns a one-to-one mapping between class transformations and labels.

### C.2 Results of Diverse Combinations of Transformations

We investigate combinations of four transformations, *i.e.*, rotation, scaling, shear, and twisting, and create unlearnable datasets using the class-wise setting. The test accuracy results across five point cloud models are presented in Tab. 7. It can be observed that the combination of rotation and scaling,

Table 7: The test accuracy (%) results obtained from training the point cloud classifiers PointNet, PointNet++, DGCNN, PointCNN, PCT using a ModelNet10 dataset generated with diverse combinations of transformations under a class-wise setting, where $\mathcal{R}$, $\mathcal{S}$, $\mathcal{H}$, and $\mathcal{W}$ correspond to rotation, scaling, shear, and twisting respectively

| Transformations | PointNet | PointNet++ | DGCNN | PointCNN | PCT | AVG |
|---|---|---|---|---|---|---|
| $\mathcal{R}$ | 46.70 | 38.99 | 49.67 | 64.87 | 27.53 | 45.55 |
| $\mathcal{S}$ | 34.80 | 57.05 | 42.62 | 33.81 | 29.63 | 39.58 |
| $\mathcal{H}$ | 64.10 | 66.41 | 71.15 | 60.68 | 60.13 | 64.49 |
| $\mathcal{W}$ | 76.98 | 54.19 | 78.63 | 86.69 | 38.55 | 67.05 |
| $\mathcal{RS}$ | 22.25 | **28.08** | **14.10** | **21.48** | **11.89** | **19.56** |
| $\mathcal{RH}$ | 30.51 | 39.98 | 45.93 | 36.23 | 33.81 | 37.29 |
| $\mathcal{RW}$ | 34.91 | 44.27 | 29.52 | 46.92 | 32.16 | 37.56 |
| $\mathcal{SH}$ | 20.26 | 44.93 | 38.66 | 43.50 | 34.80 | 36.43 |
| $\mathcal{SW}$ | 18.28 | 44.27 | 31.39 | 34.25 | 19.93 | 29.62 |
| $\mathcal{HW}$ | 63.33 | 62.56 | 61.89 | 64.76 | 51.43 | 60.79 |
| $\mathcal{RSH}$ | **15.97** | 29.19 | 36.56 | 21.81 | 31.28 | 26.96 |
| $\mathcal{RSW}$ | 25.22 | 31.50 | 23.57 | 31.83 | 38.55 | 30.13 |
| $\mathcal{SHW}$ | 21.70 | 46.37 | 55.29 | 37.11 | 38.77 | 39.85 |
| $\mathcal{RSHW}$ | 16.52 | 33.37 | 30.51 | 30.73 | 32.49 | 28.72 |

Table 8: The test accuracy (%) results with standard deviations from three runs (random seeds are set to 23, 1023, and 2023) obtained from training the point cloud classifiers PointNet, PointNet++, DGCNN, and PointCNN using a ModelNet10 dataset generated with diverse two class-wise transformations

| ModelNet10 dataset | PointNet | PointNet++ | DGCNN | PointCNN | AVG |
|---|---|---|---|---|---|
| $\mathcal{RS}$ | $15.12_{\pm6.20}$ | $26.62_{\pm5.11}$ | $25.22_{\pm10.20}$ | $17.26_{\pm3.68}$ | $21.05_{\pm0.73}$ |
| $\mathcal{RH}$ | $33.70_{\pm11.09}$ | $30.65_{\pm17.12}$ | $36.67_{\pm15.64}$ | $34.99_{\pm13.48}$ | $34.00_{\pm13.66}$ |
| $\mathcal{RW}$ | $39.83_{\pm6.71}$ | $30.18_{\pm12.22}$ | $36.71_{\pm8.10}$ | $43.47_{\pm9.20}$ | $37.55_{\pm5.07}$ |
| $\mathcal{SH}$ | $21.22_{\pm0.84}$ | $46.15_{\pm8.82}$ | $31.87_{\pm6.52}$ | $30.87_{\pm10.94}$ | $32.53_{\pm4.81}$ |
| $\mathcal{SW}$ | $23.50_{\pm6.08}$ | $51.28_{\pm6.69}$ | $38.33_{\pm6.10}$ | $28.27_{\pm5.25}$ | $35.34_{\pm2.88}$ |
| $\mathcal{HW}$ | $54.41_{\pm7.99}$ | $54.22_{\pm14.34}$ | $55.14_{\pm9.42}$ | $57.75_{\pm6.09}$ | $55.38_{\pm6.98}$ |

two types of rigid transformations, achieves the most effective unlearnable effect. The transformation parameters used in this section are consistent with Appendix B.3.

To ensure the reliability of the experimental results, we combine two different transformations and obtain the results from three runs using random seeds with 23, 1023, and 2023, then average them. The results are shown in Tab. 8. From the results on four popular point cloud models, it can be seen that when all transformations are rigid, the test accuracy is still the lowest.

## C.3    More Results of UMT Against Adaptive Attacks

To further explore the performance of UMT against adaptive attacks, we supplement the experimental results using four types of random augmentations $\mathcal{RSHW}$ in Tab. 10, and find that the conclusion is consistent with Tab. 2, *i.e.*, UMT exhibits a certain degree of robustness against the adaptive attack random transformations.

## C.4    More Results of UMT Against Semantic Segmentation

To further evaluate the UMT performance against semantic segmentation, we employ the semantic scene understanding dataset SemanticKITTI [3] in Tab. 11. It can be observed that UMT is still effective against this real-world segmentation dataset.

## C.5    Results of UMT Against SE(3) Equivariant Models

In the main text, we have discussed the robustness of UMT against rotation/scaling invariant models. The results for RIConv++ [58] (rotation invariant) and 3DGCN [26] (scaling invariant) networks in Tab. 1, as well as RIConv [57], LGR-Net [59] (rotation invariant), and 3DGCN in Tab. 4, demonstrating that these invariant networks can indeed defend against class-wise rotation and scaling. It is worth noting that these networks, which are invariant to a single transformation, cannot defend against UMT formed by a combination of multiple transformations. Thus, it appears that networks like SE(3) equivariant model, which are invariant to multiple transformations in space, can poten-

Table 9: The accuracy (%) results on clean and UMT ModelNet10 training set and test set using four point cloud classifiers. Higher accuracy values correspond to lower cross-entropy loss values.

| Training and test sets | PointNet | PointNet++ | DGCNN | PointCNN | **AVG** |
|---|---|---|---|---|---|
| Clean training set, clean test set | 89.32 | 92.95 | 92.73 | 89.54 | 91.14 |
| Clean training set, UMT test set | 55.51 | 70.93 | 74.78 | 69.71 | 67.73 |
| UMT training set, clean test set | 22.25 | 28.08 | 14.10 | 21.48 | 21.48 |
| UMT training set, UMT test set | 98.79 | 99.23 | 99.01 | 96.26 | 98.32 |

Table 10: Test accuracy (%) results using UMT training data and UMT data employing random augmentations

| ModelNet10 | PointNet | PointNet++ | DGCNN | PointCNN | AVG |
|---|---|---|---|---|---|
| Clean baseline | 89.32 | 92.95 | 92.73 | 89.54 | 91.14 |
| UMT$_{(k=4)}$ | 16.19 | 36.56 | 17.62 | 27.42 | 24.45 |
| UMT$_{(k=4)}$ + random $\mathcal{RSHW}$ | 25.99 | 61.78 | 61.89 | 44.16 | 48.46 |

tially overcome UMT. Therefore, we include experimental results for the SE(3) equivariant network SE(3)-Transformer [11] in Tab. 12.

However, it can be seen that SE(3)-Transformer cannot defend against UMT (k=3, $\mathcal{RSW}$) and UMT(k=4, $\mathcal{RSWH}$). This is because existing transformation-invariant networks, even including SE(3) invariant networks, are designed only for rigid transformations (rotation, scaling, reflection, and translation). There are currently no invariant networks proposed for non-rigid transformations like shear and twisting. Therefore, if a data protector wants UMT to be more robust, they can include non-rigid class-wise transformations to defeat existing rigid transformation-invariant networks.

## C.6  Results of Insightful Analysis

We train on clean training set and test on clean test set, train on clean training set and test on UMT test set, train on UMT training set and test on clean test set, and train on UMT training set and test on UMT test set to obtain the test accuracy results in Tab. 9 (we ensure that the UMT parameters for the UMT test set and UMT training set are consistent). Higher accuracy values indicate lower cross-entropy loss values, while lower accuracy values represent higher cross-entropy loss values. It can be observed that only when trained with the UMT training set, the loss after testing with the clean test set is high (*i.e.*, low accuracy).

## C.7  Boarder Hyper-Parameter Analysis

Additionally, we investigate the unlearnable effects across a wider range of hype-parameters $r_s, r_p, b_l, b_u$ in UMT ($k = 2$ with $\mathcal{RS}$), as shown in Tab. 13. It can be observed that our UMT scheme still exhibits a good unlearnable effect, and its key lies in the crucial role played by the class-wise setting.

## C.8  Supplementary Ablation Results

In this section, we conduct ablation experiments on the UMT scheme on the ShapeNetPart dataset (keeping experimental parameters consistent with the main experiments), as shown in Tab. 14. It can be observed that whether using only the rotation module or only the scaling module, the final unlearnable effect is not as good as UMT. This clearly demonstrates that each module contributes to the overall UMT effectiveness.

## C.9  Results of Mixture of Class-wise and Sample-wise Samples

In this section, we investigate the test accuracy results when using a mixture of different ratios of class-wise samples and sample-wise (random) samples in the dataset, as shown in Tab. 15 (experiments are conducted on the ModelNet10 dataset with 10 categories, where "2 class-wise 8 sample-wise" denotes that samples from 2 categories undergo class-wise transformations, while the remaining 8 categories undergo random transformations, and so on). We can clearly observe from the experimental results that as the proportion of class-wise transformations gradually increases, the test accuracy gradually

Table 11: **Semantic segmentation:** Evaluation of UMT on semantic segmentation task using SemanticKITTI dataset

| Evaluation metrics | Eval accuracy (%) | | mIoU(%) | |
|---|---|---|---|---|
| Segmentation models | PointNet++ | Point Transformer V2 | PointNet++ | Point Transformer V2 |
| Clean baseline | 29.89 | 72.92 | 14.16 | 54.78 |
| UMT ($k = 2, \mathcal{RS}$) | **4.69** | **19.40** | **0.80** | **13.39** |

Table 12: Test accuracy (%) results using UMT training data to train the SE(3)-Transformer

| Clean baseline | 49.07 |
|---|---|
| UMT(k=3, $\mathcal{RSW}$) | **17.51** |
| UMT(k=4, $\mathcal{RSWH}$) | **13.55** |

decreases, indicating that the unlearnable effect becomes more pronounced. This strongly suggests that the class-wise setting is more effective than the sample-wise setting.

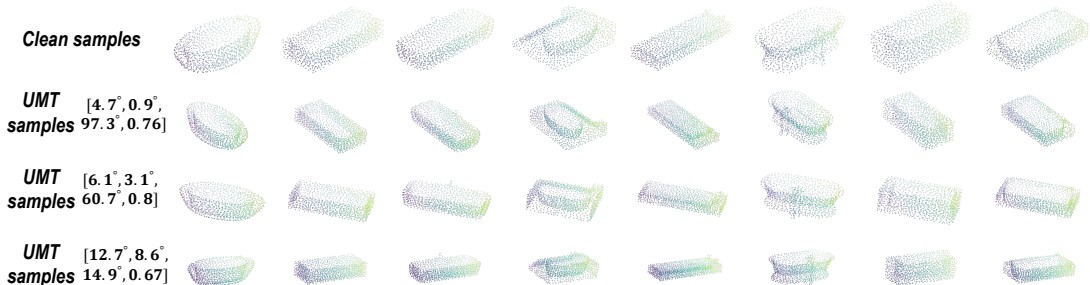

Figure 7: Clean and UMT samples from ModelNet10 dataset

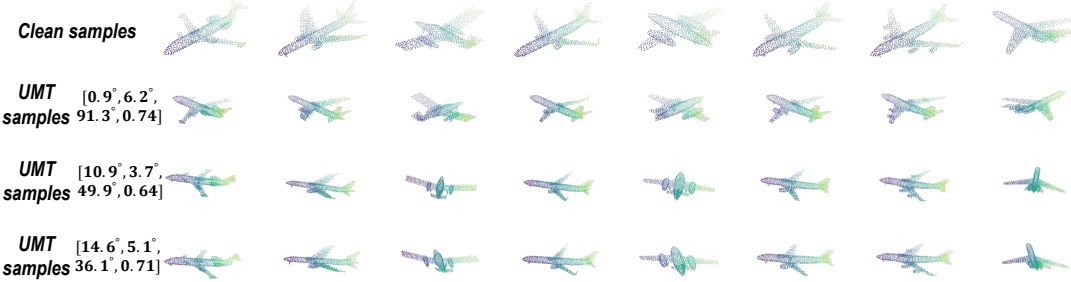

Figure 8: Clean and UMT samples from ModelNet40 dataset

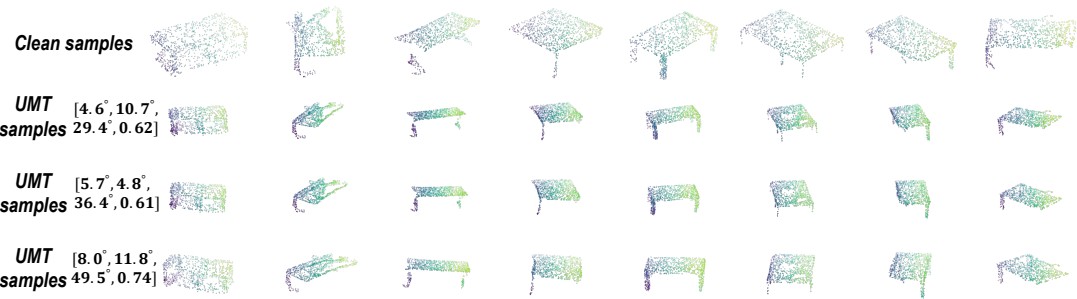

Figure 9: Clean and UMT samples from ScanObjectNN dataset

## C.10    Additional Visual Presentations for 3D Point Cloud Samples

We visualize clean point cloud samples and UMT ($k = 2$ using $\mathcal{RS}$) point cloud samples on four benchmark datasets ModelNet10 [50], ModelNet40 [50], ScanObjectNN [41], and ShapeNetPart [4],

Table 13: The test accuracy (%) results on ModelNet10 dataset with a boarder range of hype-parameters $r_s, r_p, b_l, b_u$

| $r_s, r_p, b_l, b_u$ | PointNet | DGCNN | PointCNN | AVG | $r_s, r_p, b_l, b_u$ | PointNet | DGCNN | PointCNN | AVG |
|---|---|---|---|---|---|---|---|---|---|
| 25, 120, 0.6, 0.8 | 24.78 | 23.35 | 28.19 | 25.44 | 15, 120, 0.75, 0.8 | 154 | 38.66 | 40.09 | 31.10 |
| 25, 180, 0.6, 0.8 | 26.65 | 25.99 | 20.37 | 234 | 15, 120, 0.6, 0.7 | 20.59 | 27.53 | 23.68 | 23.93 |
| 15, 90, 0.6, 0.8 | 18.39 | 20.59 | 22.03 | 20.34 | 15, 120, 0.6, 0.8 | 22.25 | 14.10 | 21.48 | 19.28 |
| 15, 240, 0.6, 0.8 | 21.26 | 30.51 | 20.70 | 24.16 | 15, 120, 0.6, 0.9 | 23.90 | 23.90 | 23.35 | 23.72 |
| 15, 120, 0.6, 1.2 | 31.50 | 28.63 | 36.78 | 32.30 | 15, 120, 0.6, 1.0 | 22.58 | 26.54 | 31.61 | 26.91 |

Table 14: **Ablation modules:** The test accuracy (%) results achieved by training on unlearnable data created by different modules on the ShapeNetPart dataset

| Benchmark datasets | Modules↓ Models⟶ | PointNet | PointNet++ | DGCNN | PointCNN | PCT | AVG |
|---|---|---|---|---|---|---|---|
| | Rotation module | 53.51 | 44.05 | 57.69 | 49.48 | **43.84** | 49.71 |
| ShapeNetPart [4] | Scaling module | 28.74 | 77.21 | 37.93 | 44.02 | 51.84 | 47.95 |
| | **UMT(k=2)** | **15.14** | **41.16** | **26.17** | **32.36** | 44.05 | **31.78** |

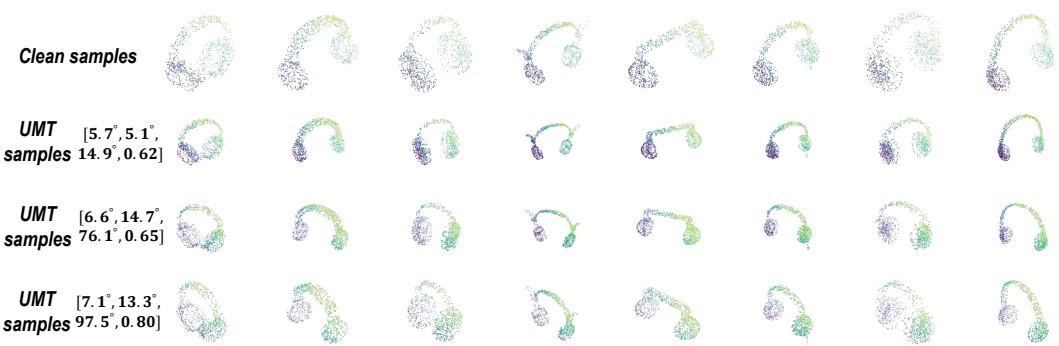

Figure 10: Clean and UMT samples from ShapeNetPart dataset

as depicted in Figs. 7 to 10, respectively. The UMT parameters consist of rotation and scaling parameters $[\alpha, \beta, \gamma, \lambda]$. It can be observed that these unlearnable samples exhibit similar feature information to normal samples, presenting good visual effects and making it difficult to be detected as abnormalities.

# D    Proofs for Theories

## D.1    Proof for Lemma 3

**Lemma 3.** *The unlearnable dataset $\mathcal{D}_u$ generated using UMT on $\mathcal{D}_c$ can also be represented using a GMM, i.e., $\mathcal{D}_u \sim \mathcal{N}(y\mathbf{T}_y\mu, \lambda_y^2 \mathbf{I})$.*

**Proof:** Assuming $y = 1$, then $\mathcal{D}_{c1} \sim \mathcal{N}(\mu, \mathbf{I})$, and we have

$$\mathbb{E}_{(x,y)\sim\mathcal{D}_{c1}} \mathbf{T}_1 x = \mathbf{T}_1 \mathbb{E}_{(x,y)\sim\mathcal{D}_{c1}} x = \mathbf{T}_1 \mu,$$
$$\mathbb{E}_{(x,y)\sim\mathcal{D}_{c1}}(\mathbf{T}_1 x - \mathbf{T}_1\mu)(\mathbf{T}_1 x - \mathbf{T}_1\mu)^\top$$
$$= \mathbf{T}_1 \mathbb{E}_{(x,y)\sim\mathcal{D}_{c1}}(x - \mu)(x - \mu)^\top \mathbf{T}_1^\top$$
$$= \mathbf{T}_1 \mathbf{I} \mathbf{T}_1^\top = \lambda_1^2 \mathcal{R}_1 \mathcal{R}_1^\top \mathbf{I} = \lambda_1^2 \mathbf{I}$$

Similarly, assuming $y = -1$, we can obtain

$$\mathbb{E}_{(x,y)\sim\mathcal{D}_{c-1}}\mathbf{T}_{-1} x = -\mathbf{T}_{-1}\mu,$$
$$\mathbb{E}_{(x,y)\sim\mathcal{D}_{c-1}}(\mathbf{T}_{-1} x - \mathbf{T}_{-1}\mu)(\mathbf{T}_{-1} x - \mathbf{T}_{-1}\mu)^\top = \lambda_{-1}^2 \mathbf{I}.$$

Thus we have: $\mathcal{D}_u \sim \mathcal{N}(y\mathbf{T}_y\mu, \lambda_y^2 \mathbf{I})$.

Table 15: **Mixture results:** The test accuracy (%) results achieved by training on the mixture data consisting of class-wise UMT samples and sample-wise UMT samples

| ModelNet10 [50] | PointNet | PointNet++ | DGCNN | PointCNN | AVG |
|---|---|---|---|---|---|
| 100% sample-wise data | 72.69 | 79.52 | 85.79 | 75.66 | 78.42 |
| 20% class-wise data, 80% sample-wise data | 65.64 | 69.05 | 78.19 | 58.04 | 67.73 |
| 40% class-wise data, 60% sample-wise data | 58.04 | 59.91 | 60.57 | 58.59 | 59.28 |
| 60% class-wise data, 80% sample-wise data | 50.11 | 48.13 | 60.35 | 46.48 | 51.27 |
| 80% class-wise data, 20% sample-wise data | 31.83 | 29.19 | 44.09 | 50.55 | 39.02 |
| 100% class-wise data | **22.25** | **28.08** | **14.10** | **21.48** | **21.48** |

## D.2 Proof for Lemma 4

**Lemma 4.** *The Bayes optimal decision boundary for classifying $\mathcal{D}_u$ is given by $P_u(x) \equiv \mathcal{A}x^\top x + \mathcal{B}^\top x + \mathcal{C} = 0$, where $\mathcal{A} = \lambda_{-1}^{-2} - \lambda_1^{-2}$, $\mathcal{B} = 2(\lambda_{-1}^{-2}\mathbf{T}_{-1} + \lambda_1^{-2}\mathbf{T}_1)\mu$, and $\mathcal{C} = \ln \frac{|\lambda_{-1}^2 \mathbf{I}|}{|\lambda_1^2 \mathbf{I}|}$.*

**Proof:** At the optimal decision boundary the probabilities of any point $x \in \mathbb{R}^d$ belonging to class $y = 1$ and $y = -1$ modeled by $\mathcal{D}_u$ are the same. Similar to the optimal decision boundary of the clean dataset $\mathcal{D}_c$, we have:

$$\frac{exp[-\frac{1}{2}(x - \mathbf{T}_{-1}\mu_{-1})^\top (\lambda_{-1}^2 \mathbf{I})^{-1}(x - \mathbf{T}_{-1}\mu_{-1})]}{\sqrt{(2\pi)^d |\lambda_{-1}^2 \mathbf{I}|}}$$

$$= \frac{exp[-\frac{1}{2}(x - \mathbf{T}_1\mu_1)^\top (\lambda_1^2 \mathbf{I})^{-1}(x - \mathbf{T}_1\mu_1)]}{\sqrt{(2\pi)^d |\lambda_1^2 \mathbf{I}|}}$$

$$\Rightarrow (x - \mathbf{T}_{-1}\mu_{-1})^\top \lambda_{-1}^{-2}(x - \mathbf{T}_{-1}\mu_{-1}) + \ln |\lambda_{-1}^2 \mathbf{I}|$$

$$= (x - \mathbf{T}_1\mu_1)^\top \lambda_1^{-2}(x - \mathbf{T}_1\mu_1) + \ln |\lambda_1^2 \mathbf{I}|$$

$$\Rightarrow \lambda_{-1}^{-2}(x^\top x - x^\top \mathbf{T}_{-1}\mu_{-1} - \mu_{-1}^\top \mathbf{T}_{-1}^\top x + \mu_{-1}^\top \mathbf{T}_{-1}^\top \mathbf{T}_{-1}\mu_{-1})$$

$$- \lambda_1^{-2}(x^\top x - x^\top \mathbf{T}_1\mu_1 - \mu_1^\top \mathbf{T}_1^\top x + \mu_1^\top \mathbf{T}_1^\top \mathbf{T}_1\mu_1) + \ln \frac{|\lambda_{-1}^2 \mathbf{I}|}{|\lambda_1^2 \mathbf{I}|} = 0$$

$$\Rightarrow (\lambda_{-1}^{-2} - \lambda_1^{-2})x^\top x - 2(\lambda_{-1}^{-2}\mu_{-1}^\top \mathbf{T}_{-1}^\top - \lambda_1^{-2}\mu_1^\top \mathbf{T}_1^\top)x$$

$$+ \mu_{-1}^\top \mu_{-1} - \mu_1^\top \mu_1 + \ln \frac{|\lambda_{-1}^2 \mathbf{I}|}{|\lambda_1^2 \mathbf{I}|} = 0$$

$$\Rightarrow P_u(x) \equiv \mathcal{A}x^\top x + 2[(\lambda_{-1}^{-2}\mathbf{T}_{-1} + \lambda_1^{-2}\mathbf{T}_1)\mu]^\top x + \ln \frac{|\lambda_{-1}^2 \mathbf{I}|}{|\lambda_1^2 \mathbf{I}|} = 0$$

$$\Rightarrow P_u(x) \equiv \mathcal{A}x^\top x + \mathcal{B}^\top x + \mathcal{C} = 0$$

where $\mathcal{A} = \lambda_{-1}^{-2} - \lambda_1^{-2}$, $\mathcal{B} = 2(\lambda_{-1}^{-2}\mathbf{T}_{-1} + \lambda_1^{-2}\mathbf{T}_1)\mu$, and $\mathcal{C} = \ln \frac{|\lambda_{-1}^2 \mathbf{I}|}{|\lambda_1^2 \mathbf{I}|}$. Besides, note that if $P_u(x)$ is less than 0, the category of the Bayesian optimal classification is -1; otherwise, it is 1.

## D.3 Proof for Lemma 5

**Lemma 5.** *Let $z \sim \mathcal{N}(0, \mathbf{I})$, $Z = z^\top z + b^\top z + c$, and $\|\cdot\|_2$ denote 2-norm of vectors. For any $t \geq 0$ and $\gamma \in \mathbb{R}$, we use Chernoff bound to have:*

$$\mathbb{P}\{Z \geq \mathbb{E}[Z] + \gamma\} \leq \frac{\exp\left\{\frac{t^2}{2(1-2t)}\|b\|_2^2 - t(\gamma + d)\right\}}{|(1 - 2t)\mathbf{I}|^{\frac{1}{2}}}$$

**Proof:** Since $\mathcal{A}$ is a constant, we have:

$$P_u(x) \equiv x^\top x + (\frac{\mathcal{B}}{\mathcal{A}})^\top x + \frac{\mathcal{C}}{\mathcal{A}} = 0$$

$$\Rightarrow P_u(x) \equiv x^\top x + b^\top x + c = 0$$

where $b = \frac{\mathcal{B}}{\mathcal{A}}, c = \frac{\mathcal{C}}{\mathcal{A}}$. Let $Z = z^\top z + b^\top z + c$ and $z \sim \mathcal{N}(0, \boldsymbol{I}) \subset \mathbb{R}^d$. Thus we have:

$$Z = z^\top z + b^\top z + c = (z^\top + \frac{1}{2}b^\top)(z + \frac{1}{2}b) + c - \frac{1}{4}b^\top b$$

$$= (z + \frac{1}{2}b)^\top (z + \frac{1}{2}b) + c - \frac{1}{4}b^\top b$$

For any $t \geq 0$ and $x \sim \mathcal{N}(0, \boldsymbol{I})$, we write the moment generating function for a quadratic random variable $Y = x^\top x$ as[5]:

$$\mathbb{E}[\exp(tY)] = \frac{1}{(2\pi)^{d/2}} \int_{\mathbb{R}^d} \exp\left\{tx^\top x\right\} \exp\left\{-\frac{1}{2}(x - \mu)^\top (x - \mu)\right\} dx$$

$$= \frac{\exp\left\{-\mu^\top \mu/2\right\}}{(2\pi)^{d/2}} \int_{\mathbb{R}^d} \exp\left\{\frac{2t - 1}{2}x^\top x + \mu^\top x\right\} dx$$

$$= \frac{\exp\left\{-\mu^\top \mu/2\right\}}{(2\pi)^{d/2}} \frac{(2\pi)^{d/2} \exp\left\{\frac{1}{2(1-2t)}\mu^\top \mu\right\}}{|\boldsymbol{I} - 2t\boldsymbol{I}|^{\frac{1}{2}}}$$

$$= \frac{\exp\left\{\frac{t}{1-2t}\mu^\top \mu\right\}}{|\boldsymbol{I} - 2t\boldsymbol{I}|^{\frac{1}{2}}}$$

$$\implies \mathbb{E}[\exp(tZ)] = \frac{\exp\left\{\frac{t}{4(1-2t)}b^\top b + t(c - \frac{1}{4}b^\top b)\right\}}{|(1 - 2t)\boldsymbol{I}|^{\frac{1}{2}}} = \frac{\exp\left\{\frac{t^2}{2(1-2t)}b^\top b + tc\right\}}{|(1 - 2t)\boldsymbol{I}|^{\frac{1}{2}}}$$

After that, we employ Chernoff bound, for some $\gamma$, we have:

$$\mathbb{P}\{Z \geq \mathbb{E}[Z] + \gamma\} \leq \frac{\mathbb{E}[\exp(tZ)]}{\exp\{t[\gamma + \mathbb{E}(Z)]\}}$$

$$= \frac{\exp\left\{\frac{t^2}{2(1-2t)}b^\top b + tc\right\}}{\exp\{t(\gamma + \mathbb{E}[z^\top z] + c)\}|(1 - 2t)\boldsymbol{I}|^{\frac{1}{2}}}$$

$$= \frac{\exp\left\{\frac{t^2}{2(1-2t)}b^\top b + tc\right\}}{\exp\{t(\gamma + Tr(\boldsymbol{I}) + \mathbb{E}(b^\top z) + c)\}|(1 - 2t)\boldsymbol{I}|^{\frac{1}{2}}}$$

$$= \frac{\exp\left\{\frac{t^2}{2(1-2t)}b^\top b + tc\right\}}{\exp\{t(\gamma + d + c)\}|(1 - 2t)\boldsymbol{I}|^{\frac{1}{2}}}$$

$$= \frac{\exp\left\{\frac{t^2}{2(1-2t)}||b||_2^2 - t(\gamma + d)\right\}}{|(1 - 2t)\boldsymbol{I}|^{\frac{1}{2}}}$$

### D.4 Proof for Theorem 6

**Theorem 6.** *For any constant $t_1$ and $t_2$ satisfying $0 \leq t_1 < \frac{1}{2}$ and $0 \leq t_2 < \frac{1}{2}$, the accuracy of the unlearnable decision boundary $P_u$ on the dataset $\mathcal{D}_c$ can be upper-bounded as:*

$$\tau_{\mathcal{D}_c}(P_u) \leq \frac{\exp\left\{\frac{t_1^2}{2(1-2t_1)}||b + 2\mu||_2^2 + t_1(\mu^\top \mu + b^\top \mu + c)\right\}}{2|(1 - 2t_1)\boldsymbol{I}|^{\frac{1}{2}}}$$

$$+ \frac{\exp\left\{\frac{t_2^2}{2(1-2t_2)}||b - 2\mu||_2^2 - t_2(\mu^\top \mu - b^\top \mu + c + 2d)\right\}}{2|(1 - 2t_2)\boldsymbol{I}|^{\frac{1}{2}}}$$

$$:= p_1 + p_2$$

---

[5][40]

*Furthermore, if $\mu^\top\mu + b^\top\mu + c + d < 0$ and $-\mu^\top\mu + b^\top\mu - c - d < 0$, we have $\tau_{\mathcal{D}_c}(P_u) < 1$. Moreover, for any $\mu \neq 0$ $\exists$ transformation matrix $\mathbf{T}_i$ such that $\tau_{\mathcal{D}_c}(P_u) < \tau_{\mathcal{D}_c}(P)$.*

**Proof:** We note that if $P_u(x)$ is less than 0, the category of the Bayesian optimal classification is -1; otherwise, it is 1. Here, $x = y\mu + z$ where $z \sim \mathcal{N}(0, \boldsymbol{I})$ and $y \in \{\pm 1\}$ since $(x, y) \sim \mathcal{D}_c$.

$$
\begin{aligned}
\tau_{\mathcal{D}_c}(P_u) &= \mathbb{E}\{\mathbb{I}(y(x^\top x + b^\top x + c) > 0)\} \\
&= \mathbb{P}\{y(\mu^\top\mu + z^\top z + 2y\mu^\top z + yb^\top\mu + b^\top z + c) > 0\} \\
&= \mathbb{P}(y = 1)\mathbb{P}\{y(\mu^\top\mu + z^\top z + 2y\mu^\top z + yb^\top\mu + b^\top z \\
&\quad + c) > 0|y = 1\} + \mathbb{P}(y = -1)\mathbb{P}\{y(\mu^\top\mu + z^\top z \\
&\quad + 2y\mu^\top z + yb^\top\mu + b^\top z + c) > 0|y = -1\} \\
&= \frac{1}{2}\mathbb{P}\{z^\top z + (b + 2\mu)^\top z + \mu^\top\mu + b^\top\mu + c > 0\} \\
&\quad + \frac{1}{2}\mathbb{P}\{-z^\top z - (b - 2\mu)^\top z - \mu^\top\mu + b^\top\mu - c > 0\} \\
&:= p_1 + p_2
\end{aligned}
$$

We can see that:

$$
\begin{aligned}
-\gamma_1 &:= \mathbb{E}\{z^\top z + (b + 2\mu)^\top z + \mu^\top\mu + b^\top\mu + c\} \\
&= Tr(\boldsymbol{I}) + \mu^\top\mu + b^\top\mu + c
\end{aligned}
$$

$$
\begin{aligned}
-\gamma_2 &:= \mathbb{E}\{-z^\top z - (b - 2\mu)^\top z - \mu^\top\mu + b^\top\mu - c\} \\
&= -Tr(\boldsymbol{I}) - \mu^\top\mu + b^\top\mu - c
\end{aligned}
$$

Applying Lemma 5, with $\gamma = \gamma_1$, $t = t_1$ for the computation of $p_1$, as well as $\gamma = \gamma_2$ and $t = t_2$ for the computation of $p_2$, where $t_1$ and $t_2$ are specific non-negative constants, we obtain:

$$
p_1 = \frac{\exp\left\{\frac{t_1^2}{2(1-2t_1)}||b + 2\mu||_2^2 + t_1(\mu^\top\mu + b^\top\mu + c)\right\}}{2|(1 - 2t_1)\boldsymbol{I}|^{\frac{1}{2}}}
$$

$$
p_2 = \frac{\exp\left\{\frac{t_2^2}{2(1-2t_2)}||b - 2\mu||_2^2 - t_2(\mu^\top\mu - b^\top\mu + c + 2d)\right\}}{2|(1 - 2t_2)\boldsymbol{I}|^{\frac{1}{2}}}
$$

This provides us with the upper bound for $\tau_{\mathcal{D}_c}(P_u)$. Nonetheless, to ensure that this upper bound is less than 1, additional conditions need to be affirmed. As $\gamma_1$ and $\gamma_2$ increase, the values of $p_1$ and $p_2$ diminish ($p_1 > 0$, $p_2 > 0$), and as $\gamma_1$ increases, $\gamma_2$ decreases (since $\gamma_1 + \gamma_2 = -2b^\top\mu$). We let $\frac{||b+2\mu||_2^2}{2}$ equal to $\alpha_1 \geq 0$, $\mu^\top\mu + b^\top\mu + c$ (also equals to $||\mu||_2^2 + c - \frac{\gamma_1 + \gamma_2}{2}$) equal to $\beta_1$, resulting in:

$$
\begin{aligned}
\frac{\mathrm{d}p_1}{\mathrm{d}t_1} &= \frac{1}{2}\frac{\mathrm{d}[\exp\{\frac{\alpha_1 t_1^2}{1-2t_1} + \beta_1 t_1\}/(1 - 2t_1)^{\frac{d}{2}}]}{\mathrm{d}t_1} \\
&= \frac{\exp\{\frac{\alpha_1 t_1^2}{1-2t_1} + \beta_1 t_1\}}{2}[(1 - 2t_1)^{-\frac{d}{2}-1}d \\
&\quad + (1 - 2t_1)^{-\frac{d}{2}}(\frac{2\alpha_1 t_1(1 - t_1)}{(1 - 2t_1)^2} + \beta_1)] \\
&= \frac{\exp\{\frac{\alpha_1 t_1^2}{1-2t_1} + \beta_1 t_1\}}{2(1 - 2t_1)^{\frac{d}{2}}}[\frac{d}{1 - 2t_1} + \frac{2\alpha_1 t_1(1 - t_1)}{(1 - 2t_1)^2} + \beta_1] \\
&= p_1(\frac{d}{1 - 2t_1} + \frac{2\alpha_1 t_1(1 - t_1)}{(1 - 2t_1)^2} + \beta_1) \\
&= \frac{2(2\beta_1 - \alpha_1)t_1^2 + 2(\alpha_1 - 2\beta_1 - d)t_1 + \beta_1 + d}{(1 - 2t_1)^2}p_1
\end{aligned}
$$

Making $|(1 - 2t_1)\boldsymbol{I}|^{\frac{1}{2}}$ meaningful requires satisfying the following condition:

$$1 - 2t_1 > 0 \Longrightarrow 0 \leq t_1 < \frac{1}{2}$$

We note that $p_1(t_1 = 0) = \frac{1}{2}$, $p_1(t_1 \to \frac{1}{2}) = +\infty$. When we set $\frac{\mathrm{d}p_1}{\mathrm{d}t_1} = 0$, we obtain that:

$$2(2\beta_1 - \alpha_1)t_1^2 + 2(\alpha_1 - 2\beta_1 - d)t_1 + d + \beta_1 = 0$$
$$\Longrightarrow t_1 = \frac{2\beta_1 - \alpha_1 + d \pm \sqrt{\alpha_1^2 - 2\alpha_1\beta_1 + d^2}}{2(2\beta_1 - \alpha_1)}$$
$$\Longrightarrow \alpha_1^2 - 2\beta_1\alpha_1 + d^2 \geq 0$$
$$\Longrightarrow \beta_1 \leq \frac{d^2 + \alpha_1^2}{2\alpha_1} = \frac{d^2}{2\alpha_1} + \frac{\alpha_1}{2}$$
$$\Longrightarrow \beta_1 \leq (\frac{d^2}{2\alpha_1} + \frac{\alpha_1}{2})_{min} = d$$

**Explanation of the Last Inequality.** Assuming we define $s(\alpha_1) = \frac{d^2}{2\alpha_1} + \frac{\alpha_1}{2}$. The minimum value of this function can be obtained using the *Arithmetic Mean-Geometric Mean Inequality* (AM-GM Inequality), which states that $a + b \geq 2\sqrt{ab}$ for $a, b \geq 0$. Thus, $s(\alpha_1) \geq 2\sqrt{\frac{d^2}{2\alpha_1} \cdot \frac{\alpha_1}{2}} = d$. Since $\beta_1 \leq s(\alpha_1)$, we can infer that $\beta_1 \leq s(\alpha_1)_{min}$, which means $\beta_1 \leq d$.

The product of the roots of the equation is: $t_{11}t_{12} = \frac{d + \beta_1}{2(2\beta_1 - \alpha_1)}$ (we assume that $t_{11} < t_{12}$). We let $f(t_1) = 2(2\beta_1 - \alpha_1)t_1^2 + 2(\alpha_1 - 2\beta_1 - d)t_1 + d + \beta_1$. Thus we have $f(0) = d + \beta_1$, $f(\frac{1}{2}) = \frac{\alpha_1}{2} > 0$.

*Situation (i):* $\beta_1 < -d$. At this time, $f(0) < 0$, $t_{11}t_{12} > 0$, based on the trend of quadratic functions and the distribution of roots, we can infer that $0 < t_{11} < \frac{1}{2}$, $t_{12} > \frac{1}{2}$, and $2\beta_1 < \alpha_1$. Thus we conclude that there exists $t_{11}$ such that $p_1(t_1 = t_{11}) < \frac{1}{2}$.

*Situation (ii):* $-d < \beta_1 < 0$. At this time, $f(0) > 0$, $t_{11}t_{12} < 0$, based on the trend of quadratic functions and the distribution of roots, we also can infer that $t_{11} < 0$, $t_{12} > \frac{1}{2}$, and $2\beta_1 < \alpha_1$. Thus we have that $p_1 \geq \frac{1}{2}$.

*Situation (iii):* $0 < \beta_1 < d$. At this time, $f(0) > 0$, the sign of $t_{11}t_{12}$ simultaneously determines the direction of the opening of the quadratic function $f(t_1)$. When $t_{11}t_{12} < 0$, $t_{11} < 0$, $t_{12} > \frac{1}{2}$, thus we have $p_1 \geq \frac{1}{2}$; when $t_{11}t_{12} > 0$, $0 < t_{11} < t_{12} < \frac{1}{2}$, the minimum point of $p_1$ is at $p_1(t_{12})$. However, it is challenging to compare $p_1(t_{12})$ and $\frac{1}{2}$ to determine which is greater or smaller.

Similarly for $p_2$, we let $\frac{||b - 2\mu||_2^2}{2}$ equal to $\alpha_2 > 0$, $-\mu^\top\mu + b^\top\mu - c - 2d$ equal to $\beta_2$, resulting in:

$$\frac{\mathrm{d}p_2}{\mathrm{d}t_2} = \frac{2(2\beta_2 - \alpha_2)t_2^2 + 2(\alpha_2 - 2\beta_2 - d)t_2 + \beta_2 + d}{(1 - 2t_2)^2}p_2$$

Thus we have the similar situation, *i.e.*, $\beta_2 < -d$. At this time, $f(0) < 0$, $t_{21}t_{22} > 0$, based on the trend of quadratic functions and the distribution of roots, we can infer that $0 < t_{21} < \frac{1}{2}$, $t_{22} > \frac{1}{2}$, and $2\beta_2 < \alpha_2$. Thus we conclude that there exists $t_{21}$ such that $p_2(t_2 = t_{21}) < \frac{1}{2}$.

Taking into account the above situations, we have that: *when $\beta_1 < -d$, $\beta_2 < -d$ (i.e., $\mu^\top\mu + b^\top\mu + c + d < 0$ and $-\mu^\top\mu + b^\top\mu - c - d < 0$), there exists $t_{11}$ and $t_{21}$ respectively, making $p_1 < \frac{1}{2}$, $p_2 < \frac{1}{2}$, i.e., $p_1 + p_2 < 1$, where $t_{11} = \frac{1}{2} + \frac{d - \sqrt{\alpha_1^2 - 2\alpha_1\beta_1 + d^2}}{2(2\beta_1 - \alpha_1)}$, $t_{21} = \frac{1}{2} + \frac{d - \sqrt{\alpha_2^2 - 2\alpha_2\beta_2 + d^2}}{2(2\beta_2 - \alpha_2)}$.*

We know that for $\mu \neq 0$, $\tau_{\mathcal{D}_c}(P) = \phi(\mu) > \frac{1}{2}$. Therefore, we need to introduce additional conditions to further ensure that $p_1 < \frac{1}{4}$, and $p_2 < \frac{1}{4}$, and consequently $\tau_{\mathcal{D}_c}(P_u) = p_1 + p_2 < \frac{1}{2} < \tau_{\mathcal{D}_c}(P)$.

To let $p_1$ satisfy $p_1 < \frac{1}{4}$ ($\alpha_1 > 0$, $\beta_1 < -d$, and $0 \le t_1 < \frac{1}{2}$), that is,

$$\exp\{\frac{\alpha_1 t_1^2}{1 - 2t_1} + \beta_1 t_1\} < \frac{(1 - 2t_1)^{\frac{d}{2}}}{2}$$

$$\Rightarrow \frac{\alpha_1 t_1^2}{1 - 2t_1} + \beta_1 t_1 < \frac{d}{2} ln(1 - 2t_1) - ln2$$

$$\Rightarrow \frac{\alpha_1 t_1^2}{1 - 2t_1} + \beta_1 t_1 - \frac{d}{2} ln(1 - 2t_1) < -ln2 = -0.693$$

We assume that $g(t) = \frac{\alpha_1 t^2}{1-2t} + \beta_1 t - \frac{d}{2} ln(1 - 2t)$. Let us assume $\alpha_1 = \frac{1}{2}$ and $d = 3$ for 3D point cloud data, then $\beta_1 < -3$. Upon analyzing the function $g(t)$, we observe that as $\beta_1$ decreases, the minimum value of $g(t)$ also decreases. We utilize various $\beta_1$ values and their corresponding function value to provide a more intuitive understanding as shown in Tab. 16.

Table 16: Different $\beta_1$ values and corresponding $g(t)$ with $t = 0.3$ and $t = 0.4$. The **bold values** represent cases where $p_1 < \frac{1}{4}$ is satisfied.

| $\beta_1$ | -4 | -6 | -8 | -10 | -12 | -14 |
|---|---|---|---|---|---|---|
| $g(0.3)$ | 0.287 | -0.313 | **-0.913** | **-1.513** | **-2.113** | **-2.713** |
| $g(0.4)$ | 1.214 | 0.414 | -0.386 | **-1.186** | **-1.986** | **-2.786** |

As for $p_2$, since $\alpha_2 \ne \alpha_1$, we need to reselect appropriate values to demonstrate the existence of $p_2 < \frac{1}{4}$. Similarly, to let $p_2$ satisfy $p_2 < \frac{1}{4}$ ($\alpha_2 > 0$, $\beta_2 < -d$, and $0 \le t_2 < \frac{1}{2}$), that is,

$$\exp\{\frac{\alpha_2 t_2^2}{1 - 2t_2} + \beta_2 t_2\} < \frac{(1 - 2t_2)^{\frac{d}{2}}}{2}$$

$$\Rightarrow \frac{\alpha_2 t_2^2}{1 - 2t_2} + \beta_2 t_2 - \frac{d}{2} ln(1 - 2t_2) < -ln2 = -0.693$$

We assume that $h(t) = \frac{\alpha_2 t^2}{1-2t} + \beta_2 t - \frac{d}{2} ln(1 - 2t)$. Let us assume $\alpha_2 = \frac{1}{3}$ and $d = 3$, similarly, we utilize various $\beta_2$ values and their corresponding function value to provide a more intuitive understanding as shown in Tab. 17.

Table 17: Different $\beta_2$ values and corresponding $h(t)$ with $t = 0.3$ and $t = 0.4$. The **bold values** represent cases where $p_2 < \frac{1}{4}$ is satisfied.

| $\beta_2$ | -4 | -6 | -8 | -10 | -12 | -14 |
|---|---|---|---|---|---|---|
| $h(0.3)$ | 0.249 | -0.351 | **-0.951** | **-1.551** | **-2.151** | **-2.751** |
| $h(0.4)$ | 1.081 | 0.281 | -0.519 | **-1.319** | **-2.119** | **-2.919** |

Therefore, we conclude that there exist $\alpha_1, \beta_1, t_1$ such that $p_1 < \frac{1}{4}$, $\alpha_2, \beta_2, t_2$ such that $p_2 < \frac{1}{4}$, *i.e.*, $p_1 + p_2 < \frac{1}{2}$. At the same time, we observe that a smaller value of $\beta$ makes it easier to satisfy the above conditions, *i.e.*, the more negative $\beta_1$ and $\beta_2$ are, the more likely it is to satisfy the above conditions. We formally combine and assert these conditions as, $b^\top \mu \ll 0$, *i.e.*, $\mu^\top \frac{\lambda_{-1}^{-2}\mathbf{T}_{-1}^\top + \lambda_1^{-2}\mathbf{T}_1^\top}{\lambda_{-1}^{-2} - \lambda_1^{-2}} \mu \ll 0$ (we sufficiently support this condition in the empirical results from Tabs. 16 and 17). Thus we conclude that *for any $\mu \ne 0$ $\exists$ transformation parameters $\mathbf{T}_i$ such that $\tau_{\mathcal{D}_c}(P_u) < \tau_{\mathcal{D}_c}(P)$.*

