# OpenReview forum: "Unlearnable 3D Point Clouds: Class-wise Transformation Is All You Need"
_NeurIPS.cc/2024/Conference — NeurIPS 2024 poster_

### Official Review · Reviewer_nf2f · 2024-07-12

**Soundness:** 3
**Presentation:** 3
**Contribution:** 2
**Rating:** 5
**Confidence:** 3

**Summary:**

The paper introduces UMT (Unlearnable Multi-Transformations), the first approach designed to render 3D point cloud data unlearnable for unauthorized deep learning models, by applying class-wise transformations. It also presents a data restoration scheme to enable authorized users to effectively train on the unlearnable data. Theoretical analysis and extensive experiments on various datasets and models demonstrate the effectiveness of UMT in safeguarding sensitive 3D data while allowing authorized access for legitimate training purposes. The main contributions include the proposal of the first 3D unlearnable scheme, a novel data restoration approach, theoretical insights into the unlearnability mechanism, and empirical validation of UMT's superiority.

**Strengths:**

1. The paper is clearly written, making complex concepts easy to understand.
2. Simple and Effective Unlearnable Scheme: The paper proposes a straightforward unlearnable scheme that leverages the characteristics of 3D data. By combining four types of 3D transformations, it effectively protects 3D point clouds data. Additionally, it allows authorized users to restore the transformed dataset using inverse transformations.
3. The extensive experiments conducted in the paper demonstrate the effectiveness of the proposed method. The UMT dataset successfully prevents unauthorized users from achieving high performance on the target test set, thereby protecting the data.
4. The method provides theoretical proof, which enhances the credibility and robustness of the approach.

**Weaknesses:**

1.	The experiments do not investigate the impact on UMT dataset performance when mixed with other datasets. Currently, transformations are applied to the entire training set, altering the distribution between the UMT training and test sets. If only part of the training set is UMT data, can the performance on the UMT test set still be maintained?
2.	The experiments rely on randomness. Are the results averaged over multiple runs to ensure reliability? While the appendix provides experiments with different random seeds, it does not show the variance of results in other experiments.
3.	The proposed transformation patterns, such as rotation and scaling, are easily detectable by comparing the UMT training set and test set. This predictability might allow unauthorized models to adapt to these transformations, potentially compromising the method's effectiveness.

**Questions:**

The questions have been listed in the Weaknesses. Please check the weaknesses.

**Limitations:**

The authors have discussed the limitations and Broader Impacts in the last section.

---

> ### Author Rebuttal · Authors · 2024-08-07
>
> > **Q1:** If only part of the training set is UMT data, can the performance still be maintained?
>
> **Re:** We are thankful for your feedback on this matter. To provide a response to this concern, we performed experiments to evaluate the impact of two UMT schemes on final test accuracy using five different UMT proportions (20%, 40%, 60%, 80%, 100%). The results are presented in Table 1 of the attached PDF.
>
> From the results, we conclude that if only part of the training set is UMT data, the test accuracy will improve, which weakens the data protector's unlearnable performance. This is because the model can only learn incorrect shortcut information from the portion of data with UMT applied, while it can learn the true knowledge from the clean data without UMT. This allows the model to achieve a certain degree of generalization on the test set. Therefore, the higher the proportion of UMT data, the better the protection effect.
>
> We appreciate the reviewer’s concern regarding the significance of different UMT data proportions. Therefore, in the revised version, we will add a section in the experimental part to discuss the effects of varying UMT data proportions.
>
>
>
>
>
>
>
>
>
> > **Q2:** Are the results averaged over multiple runs to ensure reliability?
>
> **Re:** We are deeply grateful for this valuable comment you have shared. We fully agree with the reviewer’s suggestion to perform multiple runs and average the results to improve reliability. In the main experiments presented in the paper, each result was obtained from a single run with the random seed set to 2023. To address your concerns, we conduct additional experiments on the KITTI and ScanObjectNN datasets using two more random seeds (23 and 1023). The average results with standard deviations from three runs are provided in the Table 2 and Table 3 of the attached PDF. It can be seen that UMT continues to demonstrate excellent performance. Due to space limitation, we will include the average results on remaining benchmark datasets in the revised paper.
>
>
>
> > **Q3:** Transformation patterns are easily detectable by comparing the UMT training set and test set. This predictability might allow unauthorized models to adapt to these transformations.
>
> **Re:** We greatly appreciate this valuable insight you have provided. We agree with the reviewer’s perspective that obvious transformation patterns, such as rotation and scaling, could allow unauthorized users to detect anomalies. To address this issue, our proposed *Category-Adaptive Allocation Strategy* restricts the extent of transformations to avoid excessive changes. For rotation, we limit  $\alpha$ and $\beta$ to a small range. Similarly, for scaling, shear, and twisting, we restrict the parameters controlling the transformation extent within a lower and upper bound that can be predefined by the data protector (In the revised version, we will provide a more detailed explanation of these settings and their significance). Furthermore, even if unauthorized users detect that the training data have been transformed, these UMT data may be perceived as regular augmented data because standard training procedures often use common transformations (e.g., rotation, scaling) to serve as data augmentation techniques to enhance model generalization.
>
> Moreover, let us assume that unauthorized users are aware of the UMT transformations and have designed adaptive attacks (as the reviewer has expressed concern). This situation is considered in Lines 245-249, where we discuss using random transformations as an adaptive attack against UMT, and the experimental results are presented in Table 2 (main text). The results in the last row of Table 2 demonstrate that although the effectiveness of the UMT is somewhat diminished, it still maintains a certain degree of robustness, with the test accuracy remaining 28.67% lower than the clean baseline.
>
> As for unauthorized models using networks that are invariant to these transformations for adaptive attacks, we conducted experiments with RIConv++ (a rotation-invariant network) and 3DGCN (a scale-invariant network) in Table 1, showing that these networks can successfully neutralize the effects of the transformations. We totally agree with the reviewer and acknowledge that future networks, designed to be invariant to all these types of transformations, might overcome our UMT. However, as we discuss in Section 6 (Lines 322-327), networks invariant to non-rigid transformations like twisting and shear have not yet been developed. Thus, our research calls for the creation of more transformation-invariant networks, which is also a key contribution of our work.

---

> ### Comment · Reviewer_nf2f · 2024-08-13
>
> For Q1, the experiment has a few unclear aspects. First, the dataset used in this experiment is not specified. Second, the experiment does not provide a comparison using 0% UMT data.
>
> For Q2, the experiments have addressed my concern.
>
> For Q3, I still have a concern. According to Table 3, although RIConv++ is a rotation-invariant network, the combination of several transformations is useless for RIConv++.

---

> ### Author Response · Authors · 2024-08-14
> **Response to Reviewer nf2f [1/2]**
>
> > Q1: Specify the dataset and provide a comparison using 0% UMT data.
>
> **Re:** Thank you very much for your reply. The dataset we used here is the ModelNet10 dataset. We have provided the results for 0% UMT data in the table below, and it can be seen that the trend remains consistent with our previous conclusion: the higher the proportion of UMT data, the better the protection effect.
>
> **Table X: Test accuracy (%) results when using different ratios of UMT(k=1, $\mathcal{R}$) ModelNet10 training data.**
>
> | UMT ratio | PointNet++ | PointNet | PointCNN | DGCNN |  AVG  |
> | :-------: | :--------: | :------: | :------: | :---: | :---: |
> | **100%**  |   30.51    |  29.85   |  29.46   | 35.13 | 31.24 |
> |  **80%**  |   47.03    |  47.14   |  46.92   | 48.57 | 47.42 |
> |  **60%**  |   60.35    |  62.44   |  63.55   | 62.89 | 62.31 |
> |  **40%**  |   73.79    |  66.63   |  67.40   | 69.49 | 69.33 |
> |  **20%**  |   84.03    |  84.14   |  83.26   | 85.68 | 84.28 |
> |  **0%**   |   92.95    |  89.32   |  89.54   | 92.73 | 91.14 |
>
> **Table Y: Test accuracy (%) results when using different ratios of UMT(k=2, $\mathcal{R}\mathcal{S}$) ModelNet10  training data.**
>
> | UMT ratio | PointNet++ | PointNet | PointCNN | DGCNN |  AVG  |
> | :-------: | :--------: | :------: | :------: | :---: | :---: |
> | **100%**  |   28.08    |  22.25   |  21.48   | 14.10 | 21.48 |
> |  **80%**  |   24.44    |  20.98   |  21.65   | 25.11 | 23.05 |
> |  **60%**  |   41.74    |  40.51   |  40.74   | 40.62 | 40.90 |
> |  **40%**  |   58.48    |  59.04   |  58.04   | 59.04 | 58.65 |
> |  **20%**  |   76.00    |  75.00   |  75.78   | 76.00 | 75.70 |
> |  **0%**   |   92.95    |  89.32   |  89.54   | 92.73 | 91.14 |

---

> ### Author Response · Authors · 2024-08-14
> **Response to Reviewer nf2f [2/2]**
>
> > Q3: The combination of several transformations is useless for RIConv++ on KITTI dataset.
>
> **Re:** Thank you for your thorough review. This is a really great and insightful question! After our in-depth consideration, we believe that this counter-intuitive phenomenon is caused by RIConv++'s local feature extraction process overlooking the shortcut points of UMT KITTI samples that are distant from the original sample.
>
> The following is our reasoning process: Because this phenomenon only occurs when training RIConv++ on KITTI (our KITTI dataset was created following the settings in [1,2]). Therefore, our intuition is that the unique input point cloud samples in RIConv++ requiring additional normal vectors caused this phenomenon, because RIConv++[3] claims the LRA-based normal vector is the key factor in its rotation invariance. To further investigate this, we replaced RIConv++ with PointNet (with normals) and conducted experiments on  KITTI, as shown in the table below (counterintuitive results are highlighted in bold, with "√" indicating the use of normals calculated by the LRA method in RIConv++, and "×" indicating the absence of normals).
>
> |  Datasets  |  ModelNet10  |  ModelNet40  | ShapeNetPart | ScanObjectNN  |    KITTI     |     KITTI      |     KITTI      |
> | :--------: | :----------: | :----------: | :----------: | :-----------: | :----------: | :------------: | :------------: |
> |   Models   | RIConv++ (√) | RIConv++ (√) | RIConv++ (√) | RIConv++ (√)  | RIConv++ (√) | PointNet （×） | PointNet （√） |
> |   Clean    |    86.01     |    85.82     |    97.39     | 66.34 ± 1.21  | 99.64 ± 0.09 |  98.04 ± 2.23  |  99.39 ± 0.40  |
> |  UMT (R)   |    85.16     |    81.90     |    96.96     | 62.33 ± 10.68 | 98.53 ± 1.19 | 36.23 ± 30.18  |  27.79 ± 5.81  |
> |  UMT (RS)  |    18.64     |    11.32     |     2.62     | 10.76 ± 1.76  | **99.80 ± 0.09** |  31.24 ± 7.11  |  23.83 ± 0.72  |
> | UMT (RSW)  |    13.77     |     8.63     |     3.51     | 11.69 ± 1.41  | **98.48 ± 1.61** |  19.13 ± 6.93  |  26.17 ± 6.32  |
> | UMT (RSWH) |    29.24     |    13.68     |    32.68     | 31.83 ± 2.20  | **99.34 ± 0.23** | 26.84 ± 16.26  |  29.52 ± 4.09  |
>
> We can see that: (1) Using normal vectors did not influence the final outcome of PointNet; (2) The unexpected results occurred only on the KITTI dataset. Thus, we exclude the possibility of influence from the normal vectors and shift our focus to considering the potential impacts between the KITTI samples themselves and other processes involved in the RIConv++ implementation.
>
> After carefully reading the original RIConv++ paper [3] and thoroughly inspecting the KITTI samples, we infer that this is because following the experimental procedures outlined in [1-2], the only two categories in the KITTI samples, "pedestrian" and "vehicle"(extracted from autonomous driving dataset), tend to be relatively flat in 3D space and consist of only 256 points. For example, pedestrian samples typically stand upright, with most data points concentrated around this plane-like surface, while vehicle samples are concentrated near the horizontal plane, with limited height range. Other datasets like ModelNet10 avoid this issue due to their diverse categories and 1024 points, with objects like chair and bathtub showing distinct 3D characteristics. After applying transformations to KITTI with UMT, the limited 3D sample features resulted in restricted transformation influence: the plane-like surface's features constrained the range of UMT transformations, and the number of points limited the points UMT could change, leading to local anomalies (e.g., when shear operations are applied to pedestrian samples, some points may stretch far from the plane-like surface, creating anomalies). While these anomalies would still be considered class-wise shortcuts in other models, resulting in defense effects, they are overlooked by RIConv++ since RIConv++ is designed with convolutional operators that extract local rotation-invariant features, ignoring points distant from local features.
>
> We are happy to add a discussion of this counterintuitive phenomenon in the revised manuscript and plan to delve deeper into the differing effects of UMT on the KITTI dataset and various models in future work, as this presents an intriguing area for more detailed and complete investigation. Thank you once again for your valuable feedback, we hope our response resolves your concerns. If you have any further questions, feel free to let us know. Wishing you a wonderful day :)
>
> [1] A backdoor attack against 3D point cloud classifiers. ICCV'21
>
> [2] PointCRT: Detecting backdoor in 3D point cloud via corruption robustness. MM'23
>
> [3] RIConv++: Effective rotation invariant convolutions for 3D point clouds deep learning. IJCV'22

---

### Official Review · Reviewer_Yqki · 2024-07-14

**Soundness:** 3
**Presentation:** 3
**Contribution:** 3
**Rating:** 5
**Confidence:** 2

**Summary:**

This paper studies the protection scheme against unauthorized learning on 3D point cloud data. A simple class-wise transformation method is designed to mislead the model to learn the transformation patterns of points instead of categorical knowledge. The method is evaluated on popular used 3D datasets and models.

**Strengths:**

- This paper studies a new problem and presents a reasonable and simple method to tackle the problem.

- Extensive experiments are conducted. I appreciate the comprehensive ablation studies and analyses provided in the paper.

**Weaknesses:**

- I am concerned about the value of the settings studied in the paper. Most classification datasets considered in the paper are used to develop 3D modeling algorithms and are usually far from real-world 3D/point cloud applications that require high safety standards like 3D face recognition and scene-level point clouds for autonomous driving. These scenarios usually have much more categories (e.g., facial ids) or cannot be modeled as a classification problem (e.g., scene understanding for autonomous driving). Can the proposed method be easy to transfer to these settings?

- How about the results if we randomly apply R, S, W, H when we train the model? Such strong data augmentation may lead to lower classification accuracy but the model may achieve >30% test accuracy on the UMT datasets.

- Rotation/scale/SE(3) equivariant 3D modeling has been extensively studied in previous work [r1-r4].  Will the proposed method fail if we use these models for 3D modeling?

[r1] SE(3)-Transformers: 3D Roto-Translation Equivariant Attention Networks, NeurIPS 2020

[r2] Vector neurons: A general framework for so (3)-equivariant networks, ICCV 2021

[r3] E2PN: Efficient SE(3)-Equivariant Point Network, CVPR 2023

[r4] Scale-Equivariant Deep Learning for 3D Data

**Questions:**

- The title "Class-wise Transformation Is All You Need" is quite general, which may not be helpful for readers to understand the topic studied in the paper. I was a bit confused when I first read the title along with the abstract.


Overall, this paper studies a new problem and proposes a simple approach against unauthorized learning on 3D point cloud data.  Extensive experiments and analyses are provided to evaluate the method. However, I still have concerns about the value of the settings considered in the paper and the robustness of the proposed method. I am not an expert on the topic (safety in machine learning), thus I would like to rate this paper as borderline and wait for further discussions.

**Limitations:**

The limitations and broader impacts of the method have been discussed in Section 6.

---

> ### Author Rebuttal · Authors · 2024-08-07
>
> **Response to Reviewer Yqki [1/3]**
>
> Due to space limitation, we have divided our response into three parts. Thank you for your understanding.
>
>
>
> > **Q1:** Scene understanding for autonomous driving need to be considered.
>
> **Re:** We appreciate your thoughtful and constructive feedback and strongly agree with your point about conducting experiments in more practical scenarios. We present the UMT performance on semantic segmentation task using semantic scene understanding dataset SemanticKITTI [1] in the Table R3 below. It can be observed that UMT is still effective.
>
> **Table R3:** Evaluation of UMT on semantic segmentation task using semantic scene understanding dataset SemanticKITTI [1].
>
> |      Metrics       | Eval accuracy (%) |  Eval accuracy (%)   |  mIoU (%)  |       mIoU (%)       |
> | :----------------: | :---------------: | :------------------: | :--------: | :------------------: |
> | **Semantic KITTI** |    PointNet++     | Point Transformer V2 | PointNet++ | Point Transformer V2 |
> |   Clean baseline   |       29.89       |        72.92         |   14.16    |        54.78         |
> |   **UMT (k=2)**    |     **4.69**      |      **19.40**       |  **0.80**  |      **13.39**       |
>
> Furthermore, in our original paper, we performed classification tasks on real-world datasets, such as the autonomous driving dataset KITTI and the indoor dataset ScanObjectNN (Table 1). We also performed semantic segmentation tasks of scene understanding on the indoor dataset S3DIS (Table 3). These experimental results demonstrate the effectiveness of UMT and reflect its efficacy across different tasks in real-world datasets. The key reason UMT is effective for compromising deep learning tasks across different scenarios is that it creates an erroneous mapping between the class-wise transformation shortcuts and the ground-truth labels after applying UMT to the training data. This results in a significant drop in the model's generalization performance on clean test data without any transformation shortcut (a more detailed explanation with data support can be found in Lines 125-134 of the original paper). We will add the results of SemanticKITTI in our revised paper based on your insightful suggestions.
>
> [1] *SemanticKITTI: A dataset for semantic scene understanding of LiDAR sequences.* ICCV'19
>
>
>
>
>
>
>
> > **Q2:** How about the results if we randomly apply R, S, W, H when we train the model?
>
> **Re:** We appreciate your pointing out this concern. We fully agree with your point that random data augmentation might degrade the performance of UMT. In fact, we have already reported the experimental results of using random $\mathcal{R}  \mathcal{S}$ for UMT(\mathcal{R}\mathcal{S}) in Table 2 of the original manuscript. It can be seen that after data augmentation, the test accuracy indeed increased, indicating a decrease in UMT performance. However, we also observed that the final average accuracy is still 28.67% lower than the clean baseline, demonstrating that UMT still exhibits certain robustness (as discussed in Lines 238-249). To further explore this valuable insight you provided, we supplement the experimental results using four types of random augmentations according to your suggestion in the table below, and find that the conclusion is consistent with our manuscript.
>
>
>
> **Table R4:** Test accuracy (%) results using UMT training data and UMT+random augmentation training data.
>
> |                          ModelNet10                          | PointNet | PointNet++ | DGCNN | PointCNN |  AVG  |
> | :----------------------------------------------------------: | :------: | :--------: | :---: | :------: | :---: |
> |                        Clean baseline                        |  89.32   |   92.95    | 92.73 |  89.54   | 91.14 |
> |                           UMT(k=4)                           |  16.19   |   36.56    | 17.62 |  27.42   | 24.45 |
> | UMT(k=4) + random $\mathcal{R}\mathcal{S}\mathcal{H}\mathcal{W}$ |  25.99   |   61.78    | 61.89 |  44.16   | 48.46 |
>
>
>
> While we acknowledge that these random augmentations offer some defense against UMT, and even achieve an accuracy exceeding 30%, we still believe that this cannot be considered a qualified defense. According to the prevailing view in the current literature on defenses against unlearnable attacks [1-4], a defense is only considered successful if the accuracy after the defense reaches or exceeds the level of the clean baseline. As stated earlier, random augmentations have not yet achieved this, so UMT remains robust at present.
>
> [1] *What can we learn from unlearnable datasets?*  NeurIPS'23
>
> [2] *Image shortcut squeezing: Countering perturbative availability poisons with compression.* ICML'23
>
> [3] *ECLIPSE: Expunging clean-label indiscriminate poisons via sparse diffusion purification.* ESORICS'24
>
> [4] *Purify unlearnable examples via rate-constrained variational auto-encoders.* ICML'24

---

> ### Author Response · Authors · 2024-08-07
> **Response to Reviewer Yqki [2/3]**
>
> > **Q3:** 3D face recognition also need to be considered.
>
> **Re:** We value your insightful and constructive feedback and fully concur with your suggestion to conduct experiments in more practical scenarios. We present the UMT performance on face recognition in the Table R5 below using Basel Face Model 2017 [1]  to generate 3D point cloud face scans. Our experimental setup for 3D face BFM 2017  is consistent with [2], which involves generating 100 classes of face scans (each of them contains 50 point clouds), dividing the whole 5000 face data into training and test part, containing 4000 and 1000 data respectively, and then randomly choosing 1024 points for each point clouds. It can be seen that UMT is still effective in this scenario, which is also because the class-wise transformation causes the model to overfit the transformation-based shortcut of the training data, making it difficult to generalize to clean test data.
>
> **Table R5:** Evaluation of UMT on face recognition using BFM2017-generated 3D point cloud face dataset.
>
> |     PointNet      | Clean baseline | UMT(k=1, $\mathcal{R}$) | UMT(k=2, $\mathcal{R}\mathcal{S}$) | UMT(k=3, $\mathcal{R}\mathcal{S}\mathcal{W}$) | UMT(k=4, $\mathcal{R}\mathcal{S}\mathcal{W}\mathcal{H}$) |
> | :---------------: | -------------- | ----------------------- | ---------------------------------- | --------------------------------------------- | -------------------------------------------------------- |
> | Test accuracy (%) | 98.10          | 0.81                    | 1.11                               | 0.91                                          | 1.01                                                     |
>
> [1] *Morphable face models-an open framework.* International Conference on Automatic Face & Gesture Recognition'18
>
> [2] *Toward availability attacks in 3D point clouds.* ICML'24
>
>
>
>
>
>
>
> > **Q4:** The performance of proposed method against rotation/scale/SE(3) equivariant models.
>
> **Re:** We appreciate your insight on this matter. In our original paper, we have indeed discussed the robustness of UMT against rotation/scaling invariant models. The results for RIConv++ (rotation invariant) and 3DGCN (scaling invariant) networks in Table 1, as well as RIConv, LGR-Net (rotation invariant), and 3DGCN in Table 4, demonstrating that these invariant networks can indeed defend against class-wise rotation and scaling. It is worth noting that these networks, which are invariant to a single transformation, cannot defend against UMT formed by a combination of multiple transformations. Thus, it appears that networks like SE(3), which are invariant to multiple transformations in space, can potentially overcome UMT. Therefore, following your insightful advice, we included experimental results for the SE(3) network SE (3)-Transformer [1] in the following Table R6.
>
> **Table R6:** Test accuracy results (%) of SE(3) equivariant model SE (3)-Transformer using UMT and clean training set.
>
> |                        Clean baseline                        |   49.07   |
> | :----------------------------------------------------------: | :-------: |
> |      **UMT(k=3, $\mathcal{R}\mathcal{S}\mathcal{W}$)**       | **17.51** |
> | **UMT(k=4, $\mathcal{R}\mathcal{S}\mathcal{W}\mathcal{H}$)** | **13.55** |
>
> However, it can be seen that SE (3)-Transformer cannot defend against UMT (k=3, $\mathcal{R}\mathcal{S}\mathcal{W}$) and UMT(k=4, $\mathcal{R}\mathcal{S}\mathcal{W}\mathcal{H}$). This is because existing transformation-invariant networks, even including SE(3) invariant networks [1], are designed only for rigid transformations (rotation, scaling, reflection, and translation, as shown in Fig. 7 in the original paper). There are currently no invariant networks proposed for non-rigid transformations like shear and twisting. Therefore, if a data protector wants UMT to be more robust, they can including non-rigid class-wise transformations to defeat existing rigid transformation-invariant networks. Of course, we also acknowledge that more robust invariant networks may be developed in the future, but it is exactly for this reason that the introduction of UMT will advocate for the design of more robust 3D point cloud networks, which is also where the value of our work lies.
>
>  [1] *SE(3)-Transformers: 3D Roto-Translation Equivariant Attention Networks*. NeurIPS'20

---

> ### Author Response · Authors · 2024-08-07
> **Response to Reviewer Yqki [3/3]**
>
> > **Q5:** The title is quite general, which may not be helpful for readers to understand the topic studied in the paper.
>
> **Re:** Thank you for bringing this matter to our attention. We apologize for any confusion caused by the title. To better convey the focus of the paper, we will revise the title to: "Unlearnable 3D Point Clouds: Class-wise Transformation Is All You Need".

---

> > ### Comment · Reviewer_Yqki · 2024-08-09
> >
> > Thanks for your detailed response. My concerns about data augmentation and existing SE(3)-equivariant methods have been addressed. After reading other reviews, I would like to upgrade my score to Borderline Accept.

---

> > > ### Author Response · Authors · 2024-08-09
> > > **Response to Reviewer Yqki**
> > >
> > > Dear Reviewer Yqki,
> > >
> > > We want to extend our sincere thanks for your detailed review and increasing the score during this phase. Your feedback was instrumental in helping us improve the quality of the paper, contributed significantly to the development of the community, and we are truly grateful for your support.
> > >
> > > Thank you for your time and understanding.
> > >
> > > Sincerely,
> > >
> > > Submission19662 Authors

---

### Official Review · Reviewer_BhZK · 2024-07-16

**Soundness:** 3
**Presentation:** 2
**Contribution:** 2
**Rating:** 5
**Confidence:** 4

**Summary:**

After reading the author’s response, I would like to increase my evaluation to borderline accept.

—

This paper addresses a critical issue by extending unlearnable strategies to 3D point cloud data, introducing the Unlearnable Multi-Transformations (UMT) approach. The use of a category-adaptive allocation strategy and multiple transformations is innovative and well-conceived. Notably, the paper highlights a gap in existing literature by acknowledging the challenges even authorized users face in learning from unlearnable data, proposing a data restoration scheme to address this. The theoretical and empirical validation across six datasets, sixteen models, and two tasks convincingly demonstrates the framework's effectiveness.

**Strengths:**

1. This paper studies an interesting and important problem. The problem is realistic but not widely explored.
2. The experiments of this work are comprehensive, which is admirable.

**Weaknesses:**

1. The presentation and writing need to be polished to reach the acceptance bar. For the current form, there are a series of unclear descriptions and explanations.
2. The theoretical contributions are overall weak, which provides limited insights to the research community. Besides, some theoretical analysis needs to be justified.

**Questions:**

1. The title of this paper is somewhat overclaimed or exaggerated. It is not related to the research topic of this work, which is also less informative.
2. In the introduction, the paper analyzes three issues and challenges of the research problem. However, there is no intuition about why the proposed method can handle the issues and challenges. There are just some technical details. Could the paper supplement more intuitions for a better understanding of reviewers?
3. In the introduction, there is an understanding gap between the method descriptions and theoretical analysis. Some definitions such as the decision boundary of the Bayes classifier in the Gaussian Mixture Model, are very strange, and cannot describe the work principle of the proposed method.
4. For the data protector and authorized user, do we need to add some noise to implement them with the minimax optimization? More details are needed. Besides, are they mutually reversible in practice?
5. For Eq. (3), is there some evidence about the choice of $\mathcal{A}_N$?
6. As for Property 2, why do we need all four transformation matrices we employ and the multiplicative combinations of any of these matrices are all invertible matrices? I can understand the theoretical analysis. However, it seems that this does not work for the main conclusion of this paper.
7. Similarly, the bounds in Lemma 5 and Theorem 6 are very loose. With them, it is hard to believe the proposed method can work well (although the experimental results are good). More discussions are needed.
8. The proof of Lemma 3 is simple since the space of $y$ is limited to 2. When the components of GMMs are more than 2, will the claim still hold? Could the paper add some discussions about this?
9. From Line 782 to Line 783, could the paper supplement some details to describe why the last line of inequalities holds?
10. For Line 811, the paper assumes that $\alpha_2=\frac{1}{3}$ and $d=3$. Will the assumption make the theory less general? More discussions are also needed.

---

> ### Author Rebuttal · Authors · 2024-08-07
>
> Due to space limitation, we have divided our response into three parts (including one rebuttal part and two official comments). Thank you for your understanding.
>
>
>
> #### **Response to Reviewer BhZK [1/3]**
>
> > **Q1:** The title is exaggerated, less informative, and not related to the research topic.
>
> **Re:** Thank you for pointing out that the original title may have appeared exaggerated and less informative. We agree with your assessment and have decided to revise the title to: "Unlearnable 3D Point Clouds: Class-wise Transformation Is All You Need". This new title is intended to better convey the specific nature of our research.
>
>
>
> > **Q2:** There is no intuition about why the proposed method can handle the challenges.
>
> **Re:** Thank you for pointing out the need for a clearer intuition behind our proposed method. We appreciate your feedback and would like to provide additional context to address this concern. In the introduction, we identified three main challenges: (i) Incompatibility with 3D data; (ii) Poor visual quality; (iii) Expensive time expenditure. Our intuition is that 3D transformations are custom-designed for handling 3D point cloud data (solving Challenge 1). Many of these transformations, like rotation and scaling, only alter the geometric shape without impacting visual presentation (solving Challenge 2). They are achieved through matrix operations, which are computationally linear and thus less costly than methods involving complex model optimization (solving Challenge 3). In the revised version, we will include the intuition between Line 43 and Line 44 to facilitate better understanding for the readers.
>
>
>
> > **Q3:** Some definitions such as the decision boundary of the Bayes classifier in the Gaussian Mixture Model cannot describe the work principle of the proposed method.
>
> **Re:** Thank you for your insightful review. We acknowledge that our Introduction section lacks sufficient connection and explanation between the theoretical analysis and the proposed approach, and we apologize for any understanding gap this may have caused. Here is our explanation:
>
> > To theoretically analyze UMT, we define a binary classification setup similar to that used in [1-3]. Meanwhile, we employ a Gaussian Mixture Model (GMM) to model the clean training set and use the Bayesian optimal decision boundary to model the point cloud classifier. Theoretically, we prove that the UMT training set also follows a GMM distribution and indicate the existence of that the classification accuracy of UMT dataset to be lower than that of the clean dataset in a Bayesian classifier, verifying that the proposed UMT scheme can be effective.
>
> We will insert this explanation between Lines 50 and 51 to bridge the understanding gap. We sincerely thank you once again for your valuable suggestions, which will significantly improve the quality of our paper!
>
> [1] *CUDA: Convolution-based unlearnable datasets*. CVPR'23
>
> [2] *Precise statistical analysis of classification accuracies for adversarial training.* The Annals of Statistics'22
>
> [3] *The curious case of adversarially robust models: More data can help, double descend, or hurt generalization.* Uncertainty in Artificial Intelligence'21
>
>
>
> > **Q4:** Do we need to add some noise to implement them with the minimax optimization?
>
> **Re:** Thank you for bringing this matter to our attention. We do not need to add noise for the minimax optimization. Here are our explanations: Solving the optimization problem in Eq. (1) directly is infeasible for neural networks because it necessitates unrolling the entire training procedure within the inner objective and performing backpropagation through it to execute a single step of gradient descent on the outer objective [1].  Therefore, existing unlearnable schemes to address this optimization process are generally divided into two strategies: *model-dependent noise-based optimization* [1, 2] and *model-agnostic shortcut-based operations* [3, 4]. Due to the increased computational complexity of noise-based optimization schemes for more complex point cloud data and the impact of irregular noise on sample quality, we opted for the model-agnostic shortcut-based scheme. This kind of approach usually does not involve the optimization process; however, it is necessary to activate DNN shortcuts to achieve the outcome specified in Eq. (1), like our proposed class-wise 3D transformations. The reason why our proposed model-agnostic operation can ultimately satisfy Eq. (1) is explained in Sec. 3.2 (iii). We will include a more detailed explanation of how we implement Eq. (1) at Line 89 in the revised version.
>
> [1] *Adversarial examples make strong poisons*. NeurIPS'21
>
> [2] *Unlearnable examples: Making personal data unexploitable.* ICLR'21
>
> [3] *Availability attacks create shortcuts.* KDD'22
>
> [4] *CUDA: Convolution-based unlearnable datasets*. CVPR'23
>
>
>
>
>
> > **Q5:** Are the data protector and authorized user mutually reversible in practice?
>
> **Re:** Thank you for bringing this matter to light. In practice, the optimization process between data protectors and authorized users does not necessarily need to be mutually reversible. It only requires that after the data protector releases the unlearnable data, the authorized user can use a certain method to train normally on the unlearnable data. Our proposed reversible class-wise transformation approach is just one way to achieve this goal. There are also studies that use irreversible methods to achieve this goal [1]. This paper is currently the only other work besides ours that considers authorized user access in the context of unlearnable examples. It proposes a tailored network for authorized users to learn from unlearnable image data, but this process does not involve reversible unlearnable noise. We will add more explanation about authorized users after Line 93 in the revised version.
>
> [1] *Ungeneralizable examples* CVPR'24

---

> ### Author Response · Authors · 2024-08-07
> **Response to Reviewer BhZK [2/3]**
>
> > **Q6:** Why is the choice of $\mathcal{A}_N$ for Eq. (3)?
>
> **Re:** Thank you for highlighting this issue. We set $\mathcal{A}\_N$ this way to ensure that the number of classes in the final rotation matrix is greater than or equal to N (the number of classes in the training set), thereby ensuring that the UMT scheme satisfies the class-wise setup. Concretely, in the rotation operation, each of the three directions has $\mathcal{A}\_N$ distinct angles, which means that the final rotation matrix has $\mathcal{A}\_N^3$ possible combinations. To satisfy the class-wise setup, $\mathcal{A}\_N^3$ must be at least $N$, requiring $\mathcal{A}\_N$ to be no less than $\lceil \sqrt[3]{N} \rceil$. Therefore, we configure $\mathcal{A}\_N$ in this way in Eq. (3). This point will be explained in more detail in Lines 144,145  in the revised version.
>
>
>
>
>
> > **Q7:** Why transformation matrices are all invertible matrices in Property 2?
>
> **Re:** Thank you for your careful review and for identifying this problem. The transformation matrices need to be invertible for the purpose of designing our proposed data restoration scheme (Sec. 3.4). Specifically, authorized users can build inverse matrices for the transformations after receiving the class-wise parameters from the data protector. This allows the authorized users to normally train on the protected data by leveraging the property that multiplying a matrix by its inverse results in the identity matrix (Lines 196-197). In the revised version, we will provide a detailed explanation of the necessity of Property 2 when it is mentioned for the first time in Line 159.
>
>
>
>
>
> > **Q8:** The bounds in Lemma 5 and Theorem 6 are loose.
>
> **Re:** Your feedback on this issue is greatly appreciated. We acknowledge that these bounds are quite loose. Our theorem aims to demonstrate the existence of an unlearnable situation that UMT satisfies (as stated in Line 67 and Line 189), rather than emphasizing the performance of UMT. Our theory is intended to prove that the equation $\tau\_{\mathcal{D}\_c} (P_u) < \tau\_{\mathcal{D}\_c} (P)$ has a solution, not to solve this equation. This theoretical analysis follows a previously well accepted theoretical analysis proposed by 2D unlearnable literature [1, 2]. We apologize that our theorem has given you the impression of a weak contribution, but within the current literature on unlearnable examples, this series of proofs is accepted in the theoretical analysis of the unlearnable effectiveness [1]. In the revised version, we will add a statement about the limitations of our theoretical analysis in Sec. 6.
>
> [1] *CUDA: Convolution-based unlearnable datasets*. CVPR'23
>
> [2] *Corrupting Convolution-based Unlearnable Datasets with Pixel-based Image Transformations.* arXiv:2311.18403
>
>
>
>
>
> > **Q9:** Will the claim still hold when the components of GMMs are more than 2?
>
> **Re:** We are thankful for your feedback on this matter. The claim of Lemma 3 will still hold. Here are our proofs:
>
> > **Proof (GMM with *n* components):** Assuming the dataset $\mathcal{D}\_c$ is represented by a GMM $\mathcal{N}(y\mu, \boldsymbol{I})$, where $y \in \\{y_i\\}\_{i=1}^n$. Let us take $y = y_i$, then $\mathcal{D}\_{cy_i} \sim
> > \mathcal{N}(y_i\mu, \boldsymbol{I})$, UMT data point is $(\mathbf{T}\_\{y\} x, y)$, then we have:
> >
> > **Mean:** $\mathbb{E}_ \{ (x,y) \sim  \mathcal{D}\_{cy_i} }  [\mathbf{T}\_\{y_i\} x] =  \mathbf{T}\_\{y_i\} \mathbb{E}\_\{(x,y) \sim  \mathcal{D}_{cy_i}\} x = \mathbf{T}\_\{y_i\} y_i\mu$
> >
> > **Variance:** $\mathbb{E}_ \{ (x,y) \sim  \mathcal{D}\_{cy_i} } [(\mathbf{T}\_\{y_i\} x - \mathbf{T}\_\{y_i\} y_i\mu) (\mathbf{T}\_\{y_i\} x - \mathbf{T}\_\{y_i\} y_i\mu)^{\top}] =  \mathbf{T}\_\{y_i\}      \mathbb{E}_ \{ (x,y) \sim  \mathcal{D}\_{cy_i} }  [(x - y_i\mu) (x - y_i\mu)^{\top}] \mathbf{T}\_\{y_i\}^{\top} = \mathbf{T}\_\{y_i\} \boldsymbol{I} \mathbf{T}\_\{y_i\}^{\top} = {\lambda^2\_\{y_i\} } \boldsymbol{I} $
> >
> > Thus we have: $\mathcal{D}\_u \sim \mathcal{N}(y \mathbf{T}_y \mu, \lambda\_{y}^{2} \boldsymbol{I})$.
>
> We model a binary classification problem for theoretical analysis as it allows for a simpler expression of the Bayesian decision boundary, and this setting is widely used and accepted by the community [1,2,3].
>
> [1] *CUDA: Convolution-based unlearnable datasets*. CVPR'23
>
> [2] *Precise statistical analysis of classification accuracies for adversarial training.* The Annals of Statistics'22
>
> [3] *The curious case of adversarially robust models: More data can help, double descend, or hurt generalization.* Uncertainty in Artificial Intelligence'21

---

> ### Author Response · Authors · 2024-08-07
> **Response to Reviewer BhZK [3/3]**
>
> > **Q10:** From Line 782-783, could you describe why the last line of inequalities holds?
>
> **Re:** Thank you for noting this point. Firstly, we define $\alpha_1$ as a variable $\geq 0$ on line 779. Therefore, assuming we define $s(\alpha_1) = \frac{d^2}{2\alpha_1} + \frac{\alpha_1}{2}$. The minimum value of this function can be obtained using the Arithmetic Mean-Geometric Mean Inequality (AM-GM Inequality), which states that $a + b \ge 2\sqrt{ab}$ for $a, b \ge 0$. Thus, $s(\alpha_1) \ge 2\sqrt{\frac{d^2}{2\alpha_1} \cdot \frac{\alpha_1}{2}} = d$. Since $\beta_1 \le s(\alpha_1)$, we can infer that $\beta_1 \le s(\alpha_1)_{\text{min}}$, which means $\beta_1 \le d$.  In the revised version, we will include these explanations between Line 782 and Line 783 to make the reason of this inequality more clearly.
>
>
>
>
> > **Q11:** Line 811 assumes $\alpha_2 = \frac{1}{3}, d=3$. Will this make the theory less general?
>
> **Re:** Thank you for your thorough review and for highlighting this issue. We recognize that setting specific values for these parameters may make the proof of the theorem less general, but in the special case of our existence theorem proof, this will not make Theorem 6 less general. First, since the dimensionality of the 3D point cloud data we study is 3, it is appropriate to use $d=3$ to represent the data dimension in this theoretical analysis. Secondly, assigning an appropriate value to $\alpha_2$ aims to prove the existence of a solution (as mentioned in Line 768) rather than finding the solution of the inequality $\tau\_{\mathcal{D}\_c} (P_u) < \tau_{\mathcal{D}\_c} (P)$. This similar existence theorem proof to demonstrate the effectiveness of an unlearnable scheme is also adopted by a previous unlearnable work [1]. We are very grateful for the valuable advice you have given, in the revised version, we will add the above explanation in Line 811 to make our theory more clearly.
>
> [1] *CUDA: Convolution-based unlearnable datasets.* CVPR'23

---

### Official Review · Reviewer_upBr · 2024-07-16

**Soundness:** 3
**Presentation:** 3
**Contribution:** 3
**Rating:** 7
**Confidence:** 3

**Summary:**

The paper introduces a novel approach called Unlearnable Multi-Transformations (UMT) to make 3D point cloud data unlearnable by unauthorized users. This method employs a category-adaptive allocation strategy to apply class-wise transformations, thereby preventing unauthorized training on the data. Additionally, the authors propose a data restoration scheme that allows authorized users to learn from the unlearnable data effectively. The effectiveness of UMT is validated through theoretical analysis and extensive experiments on multiple datasets and models, demonstrating its potential in protecting sensitive 3D point cloud data from unauthorized exploitation while enabling authorized access.

**Strengths:**

+ A novel unlearnable scheme specifically designed for 3D point cloud data is proposed, and it addresses the issue of enabling authorized users to utilize the protected data effectively.
+ The proposed method seems reasonable, and it has been theoretically demonstrated that the classification accuracy is lower than that of the clean dataset under the Bayes classifier's decision boundary in the Gaussian Mixture Model.
+ Extensive experiments on three synthetic and three real-world datasets, using 16 widely adopted point cloud model architectures for classification and semantic segmentation tasks, verify the superiority of the proposed method.

**Weaknesses:**

- Are object categories sensitive to the combination of transformations? For instance, are there categories where using two different sets of transformations results in significant differences in outcomes? I'm not sure if I've missed something or not, but conducting multiple experiments to observe performance and its variance might be a more reasonable approach.
- It's still not very clear how the point cloud segmentation experiments were conducted. Each object category would be in different regions of the point cloud, so would transformations be applied to the corresponding regions based on the GT? If so, would these transformations and the subsequent inverse transformations affect the segmentation results for authorized users?

**Questions:**

See the weakness

**Limitations:**

Yes, the authors mention some limitations of the proposed method, such as its vulnerability to rotation and scale transformations. However, they also note that other types of transformations are not easily compromised in the current context.

---

> ### Author Rebuttal · Authors · 2024-08-07
>
> > **Q1:** Are object categories sensitive to the combination of transformations?
>
> **Re:** Thank you for your thoughtful and valuable comment. In the original paper's Table 7, we have examined the effect of various combinations of transformations on the final performance. It can be seen that employing only rigid transformations like $\mathcal{R}$, $\mathcal{S}$, and $\mathcal{R}\mathcal{S}$ yields better results compared to using solely non-rigid transformations such as $\mathcal{H}$, $\mathcal{W}$, and $\mathcal{H}\mathcal{W}$. Nevertheless, regardless of the combination used, the final accuracy is significantly lower than the clean baseline, demonstrating excellent UMT effectiveness.
>
> Following your advice, we carried out further experiments with different random seeds (seed=2023,1023,23), and the outcomes are displayed in the Table R1 below. From the results, it can be observed that when k=2, the average performance across different combinations are not significantly different, all exhibiting good unlearnable effects of UMT. Additionally, the combination of only rigid transformations $\mathcal{R}\mathcal{S}$ outperforms that using only non-rigid transformations $\mathcal{H}\mathcal{W}$, with mixed combinations of both types of transformations yielding intermediate results, which aligns with the findings previously discussed and presented in Table 7. Thank you once again for your valuable advice. We will include a discussion section in our experiments to explore this intriguing phenomenon.
>
>
>
> **Table R1:** Average test accuracy (%) results (from three runs with random seeds 23, 1023, 2023) using diverse combinations of transformation for UMT.
>
> |        ModelNet10         |   PointNet    |  PointNet++   |     DGCNN     |   PointCNN    |      AVG      |
> | :-----------------------: | :-----------: | :-----------: | :-----------: | :-----------: | :-----------: |
> | $\mathcal{R} \mathcal{S}$ | 15.12 ± 6.20  | 26.62 ± 5.11  | 25.22 ± 10.20 | 17.26 ± 3.68  | 21.05 ± 0.73  |
> | $\mathcal{R} \mathcal{H}$ | 33.70 ± 11.09 | 30.65 ± 17.12 | 36.67 ± 15.64 | 34.99 ± 13.48 | 34.00 ± 13.66 |
> | $\mathcal{R} \mathcal{W}$ | 39.83 ± 6.71  | 30.18 ± 12.22 | 36.71 ± 8.10  | 43.47 ± 9.20  | 37.55 ± 5.07  |
> | $\mathcal{S} \mathcal{H}$ | 21.22 ± 0.84  | 46.15 ± 8.82  | 31.87 ± 6.52  | 30.87 ± 10.94 | 32.53 ± 4.81  |
> | $\mathcal{S} \mathcal{W}$ | 23.50 ± 6.08  | 51.28 ± 6.69  | 38.33 ± 6.10  | 28.27 ± 5.25  | 35.34 ± 2.88  |
> | $\mathcal{H} \mathcal{W}$ | 54.41 ± 7.99  | 54.22 ± 14.34 | 55.14 ± 9.42  | 57.75 ± 6.09  | 55.38 ± 6.98  |
>
>
>
> > **Q2:** How the point cloud segmentation experiments were conducted?
>
> **Re:** We appreciate your valuable comments regarding this issue. Yes, you are correct. In the semantic segmentation scenario, we use class-wise transformations based on the ground-truth labels of the corresponding regions. We will clarify this process in Sec. 3.3.1 (methodology part) and Sec. 4.2 (experimental part) in the updated version.
>
>
>
> > **Q3:** Would the transformations and the subsequent inverse transformations affect the segmentation results for authorized users?
>
> **Re:** We appreciate your pointing out this concern. The transformations and the subsequent inverse transformations do not negatively affect the final segmentation results for authorized users. After applying UMT and then adding the inverse transformation of the restoration scheme, the experimental results are shown in the Table R2 below (each result is the average of three runs to ensure reliability). It can be seen that the final results after applying UMT + Restoration scheme are marginally above the clean baseline results. The UMT+data restoration scheme did not affect the final segmentation results because the data restoration scheme is designed to break UMT's class-wise transformation patterns through class-wise inverse transformations of reversible matrices. The original matrix's effect is neutralized when the class-wise reversible matrix is multiplied by the corresponding class-wise inverse matrix, thus eliminating the unlearnable effects of UMT, allowing authorized users' point cloud DNNs to learn the features of samples, thereby achieving standard segmentation performance. This principle is consistent with its effectiveness in classification tasks.
>
> **Table R2: Semantic segmentation.** Test accuracy (%) and mIoU (%) results using standard training and UMT+data restoration scheme.
>
> |     Test accuracy  (%)     |    PointNet++    |   Point Transformer v3   |      SegNN       |       AVG        |
> | :------------------------: | :--------------: | :----------------------: | :--------------: | :--------------: |
> |       Clean baseline       |      74.76       |          74.72           |      79.00       |      76.16       |
> | **UMT+Restoration scheme** | **80.23 ± 2.42** |     **76.86 ± 1.29**     | **80.14 ± 0.31** | **79.08 ± 1.23** |
> |        **mIoU (%)**        |  **PointNet++**  | **Point Transformer v3** |    **SegNN**     |     **AVG**      |
> |       Clean baseline       |      40.06       |          40.57           |      50.27       |      43.63       |
> | **UMT+Restoration scheme** | **48.61 ± 3.24** |     **43.28 ± 1.35**     | **50.37 ± 0.06** | **47.42 ± 1.43** |

---

### Author Rebuttal · Authors · 2024-08-07

## **Global Response**

We express our heartfelt thanks to all the reviewers for their valuable time and are encouraged that they found the paper to be:

1. **Clearly written, making complex concepts easy to understand** *(nf2f)*.
2. The studied problem is **interesting** *(BhZK)*, **important** *(BhZK)*, **realistic** *(BhZK)*, and **new** *(Yqki)*.
3. The proposed scheme is **novel** *(upBr)*, **reasonable** *(upBr, Yqki)*, **simple** *(Yqki, nf2f)*, and **effective** *(nf2f)*.
4. **Theoretical proofs** enhance the **reasonableness** *(upBr)*, **credibility** *(nf2f)*, and **robustness** *(nf2f)* of the scheme.
5. **Extensive experiments** verify the effectiveness of the proposed method *(upBr, BhZK, Yqki, nf2f)*.

We have answered the reviewers' concerns and questions in response to their official reviews and are open to discussing any additional issues that may arise. We would greatly appreciate any further feedback on our detailed rebuttal. Due to the space limitation of each rebuttal, we include tables of some experimental results addressing your concerns in the **attached PDF file**.

---

### Comment · Area_Chair_y9qA · 2024-08-12
**Help Check the Rebuttal, Make Discussions, and Update the Final Recommendations**

Dear Reviewers,

Thanks for serving as a reviewer for the NeurIPS. The rebuttal deadline has just passed.

The author has provided the rebuttal, could you help check the rebuttal and other fellow reviewers' comments, make necessary discussions, and update your final recommendations as soon as possible?

Thank you very much.

Best,

Area Chair

---

### Decision · Program_Chairs · 2024-09-25

**Decision:**

Accept (poster)

**Comment:**

This paper was reviewed by four experts in the field. The recommendations are (Accept, Borderline Accept x 3). Based on the reviewers' feedback, the decision is to recommend the acceptance of the paper. The reviewers did raise some valuable concerns (especially the **overclaimed and exaggerated** paper title by Reviewers BhZK and Yqki, missed detailed and convincing experimental evaluations by Reviewers Yqki and nf2f) that should be addressed in the final camera-ready version of the paper. The authors are encouraged to make the necessary changes to the best of their ability.